# Quantum order-by-disorder induced phase transition in Rydberg ladders with staggered detuning

Madhumita Sarkar, Mainak Pal, Arnab Sen, and K. Sengupta

School of Physical Sciences, Indian Association for the Cultivation of Science, Kolkata 700032, India
* ksengupta1@gmail.com

August 9, 2022

## Abstract

$^{87}$Rb atoms are known to have long-lived Rydberg excited states with controllable excitation amplitude (detuning) and strong repulsive van der Waals interaction $V_{\mathbf{rr'}}$ between excited atoms at sites $\mathbf{r}$ and $\mathbf{r'}$. Here we study such atoms in a two-leg ladder geometry in the presence of both staggered and uniform detuning with amplitudes $\Delta$ and $\lambda$ respectively. We show that when $V_{\mathbf{rr'}} \gg (\ll)\Delta, \lambda$ for $|\mathbf{r} - \mathbf{r'}| = 1(> 1)$, these ladders host a plateau for a wide range of $\lambda/\Delta$ where the ground states are selected by a quantum order-by-disorder mechanism from a macroscopically degenerate manifold of Fock states with fixed Rydberg excitation density $1/4$. Our study further unravels the presence of an emergent Ising transition stabilized via the order-by-disorder mechanism inside the plateau. We identify the competing terms responsible for the transition and estimate a critical detuning $\lambda_c/\Delta = 1/3$ which agrees well with exact-diagonalization based numerical studies. We also study the fate of this transition for a realistic interaction potential $V_{\mathbf{rr'}} = V_0/|\mathbf{r} - \mathbf{r'}|^6$, demonstrate that it survives for a wide range of $V_0$, and provide analytic estimate of $\lambda_c$ as a function of $V_0$. This allows for the possibility of a direct verification of this transition in standard experiments which we discuss.

# 1 Introduction

It is usually expected that fluctuations, thermal or quantum, in a generic many-body system shall lead to suppression of order. However, a somewhat less intuitive counterexample, namely, stabilization of order in a system with competing interactions due to quantum or thermal fluctuations is now well-accepted [1–3]. This phenomenon occurs due to the presence of macroscopically degenerate manifold of classical ground states in such systems; the presence of fluctuations then may lift this degeneracy leading to a ground state with definite order. This mechanism is dubbed as order-by-disorder [1]. Examples of this phenomenon is seen in a variety of quantum many-body systems involving spins [4–6], bosons [7–9] and fermions [10].

In recent years, ultracold atoms in optical lattice have proved to be efficient emulators of several well-known model Hamiltonians in condensed matter systems [11]. A primary example of this is the emulation of the Bose-Hubbard model [12–15] using ultracold bosonic atoms [16]. A study of such bosons in their Mott phases and in the presence of a tilted lattice generating an artificial electric field have led to the realization of a translational symmetry broken ground state and an associated quantum phase transition which belongs to the Ising universality class [17–24]. More recently, another such ultracold atom system, namely, a chain of $^{87}$Rb atoms supporting long-lived Rydberg excited states has also been studied in detail. The amplitude of realizing a Rydberg excitations at any site of such a chain, termed as detuning, can be individually controlled using external Raman lasers [25–28]. These atoms, in their Rydberg excited states, experience strong repulsive van der Waals(vdW) interaction leading to a finite Rydberg blockade radius [29–33]. Such a system leads to the realization of symmetry broken phases separated by both Ising and non-Ising quantum phase transitions [25, 26, 34–37]. The out-of-equilibrium dynamics of such systems leading to realization of the central role of quantum scars in such dynamics have also been studied both theoretically [38–43] and experimentally [26]. These studies have also been extended to two-dimensional (2D) arrays of Rydberg atoms leading to the possibility of realization of Kitaev spin liquids in these systems [44–49].

In all of the above-mentioned studies [25–28, 34–49], the detuning of the Rydberg atoms have been assumed to be uniform. However, recently, chains of such Rydberg atoms in the presence of an additional staggered detuning have also been studied extensively [50–59]. The main motivation behind such studies stemmed from the possibility of realization of quantum link models using ultracold atom systems [50–53, 55, 56]; these models are well-known to exhibit quantum confinement [60, 61] which is usually difficult to realize in a generic condensed matter setup [57, 58, 62]. More recently, the out-of-equilibrium dynamics of such models have also been studied; it was found that they host several interesting phenomena such as ultraslow dynamics following a quench [57], dynamical freezing in the presence of a periodic drive, and Floquet scars [59]. However, to the best of our knowledge, the phases of coupled ladders of Rydberg atoms with staggered detuning have not been studied in detail so far.

In this work, we study the phases of such coupled chains of Rydberg atoms in the presence of both uniform and staggered detunings with amplitudes $\lambda$ and $\Delta$ respectively. In addition, such a system allows for a coupling between the Rydberg excited and the ground states of the atoms with an amplitude $w$ [25, 26]. Such coupled chains leads to a two-leg ladder geometry as shown in Fig. 1(a). The corresponding blockade radius for the Rydberg excitations, which prevents existence of two Rydberg excitation on nearest-neighbor sites in such ladders, is also shown in Fig. 1(a). In our work, we first concentrate in the regime where the vdW interaction between the Rydberg atoms satisfy $V_{\mathbf{rr}'} \ll \lambda, \Delta, w$ outside the blockade radius. In this regime, we find that for a wide range of $\lambda/\Delta$ such a system shows a plateau where the Rydberg excitation density, $n$, is fixed to $n = 1/4$. The plateau destabilizes at $\lambda/\Delta \simeq \pm 1$ leading to a change in $n$; we analyze this behavior using a variational wavefunction method which agrees well with exact diagonalization (ED) results.

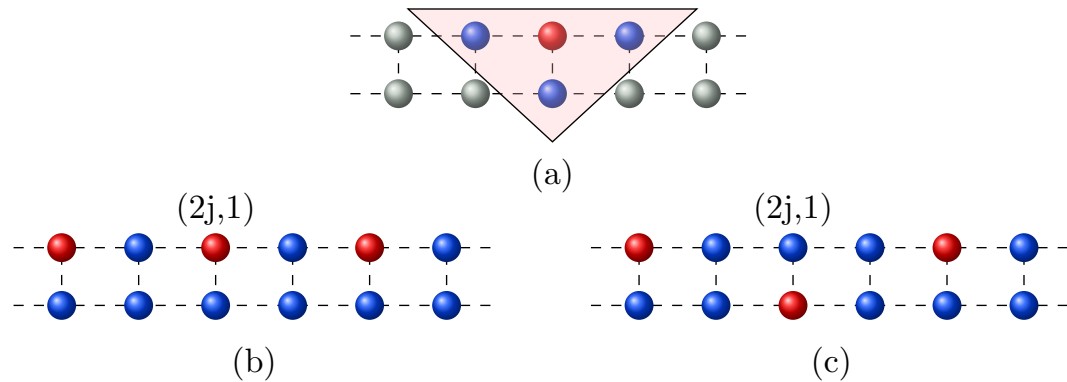

Figure 1: (a) Schematic representation of a two-leg Rydberg ladder. The circles represent sites on the Rydberg chain. The presence of a Rydberg excitation on a site is denoted by the red circle; this precludes excitations of nearby sites (denoted by blue circles). The corresponding Rydberg blockade radius is schematically shown by the triangle. The other sites (grey circles) may have Rydberg excitations since they would experience weaker repulsive vdW interaction falling off as $1/|\vec{r}|^6$ with distance $|\vec{r}|$ from the site with a Rydberg excited atom (red circle). (b) Schematic representation of one of the states with Rydberg excitation density $n = 1/4$ with excitations occurring at even sites of one of the ladders. This state is one of the two possible classical Fock states which represents the ground state for $\lambda > \lambda_c$. (c) A similar diagram for a state in the low-energy manifold with $n = 1/4$ whose superposition forms the ground state at $\lambda < \lambda_c$ but not for $\lambda > \lambda_c$. See text for details.

The ground state of the system within the plateau involves macroscopic number of degenerate classical Fock states having a Rydberg excitation in one of the two sites in every even rung of the ladder (Fig. 1(b) and 1(c)) and is chosen by a quantum order-by-disorder mechanism from these classical Fock states. For $\lambda > \lambda_c$, the ground state breaks $\mathbb{Z}_2$ symmetry; it consists of Fock states where all Rydberg excitations are localized in any one of the two rungs of the ladder (Fig. 1(b)). In contrast, for $\lambda < \lambda_c$, we find a unique ground state which constitutes a macroscopic superposition of Fock states with fixed $n = 1/4$. One such state is schematically shown in Fig. 1(c). This necessitates the presence of a quantum phase transition which belongs to the 2D Ising universality class at $\lambda = \lambda_c$. We carry out a Schrieffer-Wolff transformation to obtain an effective low-energy Hamiltonian which provides analytical insight into this transition and identifies the competing terms responsible for it. Our perturbative analysis, based on the effective Hamiltonian, estimates the critical point to be $\lambda_c \simeq \Delta/3$ which shows an excellent match with exact numerical results based on ED. Thus our analysis indicates that the order-by-disorder mechanism necessarily leads to a quantum phase transition separating the above-mentioned ground states. As we detail in the appendix, our study naturally leads to a class of spin models involving ladders with $\ell_0$ legs that realize ground states with broken $\mathbb{Z}_{\ell_0}$ symmetry. For $\ell_0 = 3$, as we show both using ED and analytic perturbation theory in the appendix, this provides a route to realization of a quantum critical point belonging to the three-state Potts universality class in 2D. To the best our knowledge, such emergent transitions arising from an order-by-disorder mechanism have not been reported in the context of Rydberg systems in the literature [25–28, 34–37, 44–49].

Next, in an attempt to make contact with realistic experimental systems, we discuss the fate of this transition for the two-leg ladder in the presence of a realistic vdW interaction characterized by $V_{\mathbf{rr'}} = V_0/|\mathbf{r} - \mathbf{r'}|^6$. We discuss an experimentally relevant regime where $V_0 \gg \lambda, \Delta, w$ so that states with one or more nearest-neighbor Rydberg excitations do not

feature in the ground state manifold. However, Rydberg excitations on second and higher neighboring sites now experience a finite repulsive interaction. We show that the main effect of having such a second and higher neighbor interaction is to shift $\lambda_c/\Delta$ to a higher value; the Ising transition still persists for a wide range of $V_0$ and $w$. We analyze the transition using a Van Vleck perturbation theory, discuss the significance of the order-by-disorder mechanism for its stability, and provide an estimate of $\lambda_c$ as a function of $V_0$ and $w$. Finally, we discuss experiments which can test our theory.

The plan of the rest of the paper is as follows. In Sec. 2, we define the model Hamiltonian for Rydberg atoms on a two-leg ladder. Next, in Sec. 3, we analyze the phases of the model by ignoring the second and higher neighboring repulsive interaction between the Rydberg atoms. This is followed by Sec. 4 where we discuss the effect of finite $V_0$. Finally, we discuss our main results, point out possible experiments which can test our theory, and conclude in Sec. 5. We discuss our variational wavefunction results around $\lambda \simeq \Delta$ in App. A and chart out the phases of ladders with $\ell_0 > 2$ rungs, which could be more challenging to realize within current experimental setups, in App. B.

## 2 Model Hamiltonian

In this section, we outline the model Hamiltonian for the Rydberg atoms. We consider an arrangement of two Rydberg chains each having $2L$ sites shown schematically in Fig. 1(a). The sites of these chains host $N = 4L$ Rydberg atoms whose low-energy effective Hamiltonian is written as [25–28]

$$H = -\sum_{j=1}^{2L}\sum_{\ell=1}^{2}\left(w\sigma_{j,\ell}^{x} + \frac{1}{2}[\Delta(-1)^{j} + \lambda]\sigma_{j,\ell}^{z}\right) + \sum_{\mathbf{r}\neq\mathbf{r}'}V_{\mathbf{r}\mathbf{r}'}\hat{n}_{\mathbf{r}}\hat{n}_{\mathbf{r}'}, \qquad (1)$$

where $\sigma_{j,\ell}^{\alpha}$ for $\alpha = x, y, z$ denotes the usual Pauli matrices on the site $j$ of the $\ell^{\text{th}}$ chain and the lattice spacing $a$ between the sites is set to unity so that the coordinate $\mathbf{r}$ of any site is denoted by integers $j$ and $\ell$: $\mathbf{r} = (j, \ell)$. Here $\lambda$ and $\Delta$ denote the amplitudes of uniform and staggered detuning respectively, and $w > 0$ denotes the coupling strength between the Rydberg ground and excited states. We note at the outset that the choice of the sign of $w$ shall not affect the results obtained in this work. Thus for any site with coordinate $\mathbf{r}$,

$$\sigma_{\mathbf{r}}^{x} \equiv \sigma_{j,\ell}^{x} = (|G\rangle_{\mathbf{r}\,\mathbf{r}}\langle R| + \text{h.c.}), \quad \hat{n}_{\mathbf{r}} \equiv \hat{n}_{j,\ell} = (1 + \sigma_{j,\ell}^{z})/2, \qquad (2)$$

where $|R\rangle_{\mathbf{r}} \equiv |\uparrow_{\mathbf{r}}\rangle$ and $|G\rangle_{\mathbf{r}} \equiv |\downarrow_{\mathbf{r}}\rangle$ denotes excited and ground states of a Rydberg atom on the site $\mathbf{r}$ respectively and $\hat{n}_{\mathbf{r}}$ is the number operator corresponding to the Rydberg excitations. In the Rydberg excited state, the atoms experience strong vdW repulsion which is modeled by $V_{\mathbf{r}\mathbf{r}'}$ given by

$$V_{\mathbf{r}\mathbf{r}'} = V_0/|\mathbf{r} - \mathbf{r}'|^{6}, \qquad (3)$$

where $V_0$ is the interaction strength. It is well-known that in a typical experiment, $V_0$ can be tuned to induce Rydberg blockade between neighboring sites. For the rest of this paper, we shall assume that $V_0 \gg \Delta_0, \lambda, w$ so that the neighboring sites of the two-rung ladder have at most one Rydberg excited atom (Fig. 1(a)). In this limit, one can split the Hamiltonian in two parts $H_a$ and $H_b$ given by

$$H_a = \sum_{\mathbf{r},\mathbf{r}'}V_0\hat{n}_{\mathbf{r}}\hat{n}_{\mathbf{r}'}\delta_{|\mathbf{r}-\mathbf{r}'|-1},$$

$$H_b = -\sum_{j=1}^{2L}\sum_{\ell=1}^{2}\left(w\sigma_{j,\ell}^{x} + \frac{1}{2}[\Delta(-1)^{j} + \lambda]\sigma_{j,\ell}^{z}\right) + \frac{V_0}{2}\sum_{\mathbf{r}\mathbf{r}'}\frac{n_{\mathbf{r}}\hat{n}_{\mathbf{r}'}}{|\mathbf{r}-\mathbf{r}'|^{6}}(1 - \delta_{|\mathbf{r}-\mathbf{r}'|-1}). \qquad (4)$$

In the limit, where $V_0$ is the largest energy scale in the problem, it is possible to define a hierarchy of energies for the eigenstates based on the number of nearest-neighbor Rydberg excitations [27]. This can be encoded via a projection operator $T_m$ which satisfies $[H_a, T_m] = mV_0 T_m$. The role of $T_m$ is to project the Hamiltonian into a sector of $m$-nearest neighbor Rydberg excitation. In what follows, we shall be concerned with the effective Hamiltonian in the sector of $m = 0$ nearest-neighbor dipoles. A straightforward calculation similar to the one carried out in Ref. [27] for a Rydberg chain yields [17, 63]

$$
\begin{aligned}
H_{\text{eff}}^V &= -\sum_{j=1}^{2L}\sum_{\ell=1}^{2} w\tilde{\sigma}_{j,\ell}^x - \sum_{j=1}^{2L}\sum_{\ell=1}^{2}\frac{1}{2}[\Delta(-1)^j + \lambda]\sigma_{j,\ell}^z + \frac{V_0}{2}\sum_{\mathbf{rr}'}\frac{n_{\mathbf{r}}\hat{n}_{\mathbf{r}'}}{|\mathbf{r}-\mathbf{r}'|^6}(1 - \delta_{|\mathbf{r}-\mathbf{r}'|-1}), \\
\tilde{\sigma}_{j,\ell}^x &= P_{j-1,\ell}\left(\prod_{\ell'\neq\ell}P_{j\ell'}\sigma_{j,\ell}^x\right)P_{j+1,\ell},
\end{aligned} \tag{5}
$$

where we have used the fact that $H_0$ does not contribute in this sector and $P_{j,\ell} = (1 - \sigma_{j,\ell}^z)/2$ is the local projection operator which ensures the absence of nearest-neighbor Rydberg excitations. The higher-order terms can be systematically computed involving $m \neq 0$ nearest-neighbor Rydberg excitations but are unimportant for the regime that we are interested. Furthermore, in the regime where $V_0 \gg \lambda, \Delta \gg w \gg V_0/(\sqrt{2})^6$, it is possible to neglect the last term in $H_{\text{eff}}$. In this regime one obtains the following generalized PXP Hamiltonian [17, 63] on the 2-leg ladder

$$
H_{\text{eff}} = -\sum_{j=1}^{2L}\sum_{\ell=1}^{2} w\tilde{\sigma}_{j,\ell}^x - \frac{1}{2}\sum_{j=1}^{2L}\sum_{\ell=1}^{2}[\Delta(-1)^j + \lambda]\sigma_{j,\ell}^z. \tag{6}
$$

In the next section, we shall obtain the ground state phase diagram of $H_{\text{eff}}$ given in Eq. 6. The effect of finite second and higher-neighboring interaction shall be discussed, using $H_{\text{eff}}^V$ (Eq. 5) in Sec. 4.

## 3 Phases of $H_{\text{eff}}$

To understand the phase-diagram of $H_{\text{eff}}$, we first set $w = 0$. It is then easy to see that for $\lambda < 0$ and $|\lambda| \gg \Delta$, we have $\langle\sigma_{j,\ell}^z\rangle = -1$ on all sites leading to $n = \langle\sum_{\vec{r}}\hat{n}_{\vec{r}}\rangle/N = 0$. Similarly for $\lambda > 0$ and $\lambda \gg \Delta$, the ground states hosts maximal number of possible up-spins or Rydberg excitations. However, due to the constraint of having at most one Rydberg excitation on neighboring sites, it can have such excitations only on $N/2$ sites. This results in a net Rydberg excitation density of $n = 1/2$ and leads to a two-fold degenerate ground state. In between for $-\Delta \leq \lambda \leq \Delta$, the ground state hosts one Rydberg excitation on every even rung of the ladder and thus has $n = 1/4$. This is due to the fact that such excitations requires $\delta E_{\text{even}} = -(\lambda + \Delta) < 0$ in this regime. In contrast, a Rydberg excitation on an odd site costs $\delta E_{\text{odd}} = \Delta - \lambda > 0$. The ground state manifold therefore has macroscopic $2^L$ fold classical degeneracy for $w = 0$. At $\lambda = \pm\Delta$, $n$ exhibits jumps for $w = 0$ which constitute first order transitions.

These discontinuous transitions become smooth crossovers due to quantum fluctuations introduced by a finite $w$ (Fig. 2 (a)). This can be understood using a variational wavefunction based analysis. Here we illustrate this for $\lambda/\Delta \simeq -1$; a similar analysis can be carried out for $\lambda \simeq \Delta$ and is shown in App. A. We start by noting that for $|\lambda| \gg \Delta, w$ and $\lambda < 0$, the ground

state of $H_{\text{eff}}$ is given by

$$|\psi_1\rangle = \prod_{j=1}^{L} |\downarrow_{2j-1,1}\downarrow_{2j-1,2}; \downarrow_{2j,1}\downarrow_{2j,2}\rangle. \tag{7}$$

In contrast, for $\Delta > |\lambda|$, it is clearly energetically favorable to have a Rydberg excitation on even sites. A variational wavefunction representing such a state is given by

$$|\psi_2\rangle = \prod_{j=1}^{L} \big(\cos\phi|\downarrow_{2j-1,1}\downarrow_{2j-1,2}; \uparrow_{2j,1}\downarrow_{2j,2}\rangle + \sin\phi|\downarrow_{2j-1,1}\downarrow_{2j-1,2}; \downarrow_{2j,1}\uparrow_{2j,2}\rangle\big). \tag{8}$$

Note that we have chosen the variational parameter $\phi$ to be independent of position since we intend to carry out a mean-field analysis of the problem here. Near $\lambda/\Delta = -1$, we construct a variational wavefunction $|\psi_v\rangle = \cos\theta|\psi_1\rangle + \sin\theta|\psi_2(\phi)\rangle$ which leads to $E_v = \langle\psi_v|H|\psi_v\rangle$ given by

$$E_v = (\lambda\cos^2\theta - \Delta\sin^2\theta) - \sqrt{2}w\sin2\theta\cos(\phi - \pi/4), \tag{9}$$

where we have ignored an irrelevant constant term. The minimization of $E_v$ fixes

$$\phi = \phi_0 = \pi/4, \quad \theta = \theta_0 = \frac{1}{2}\arctan\left[\frac{-2\sqrt{2}w}{\lambda + \Delta}\right]. \tag{10}$$

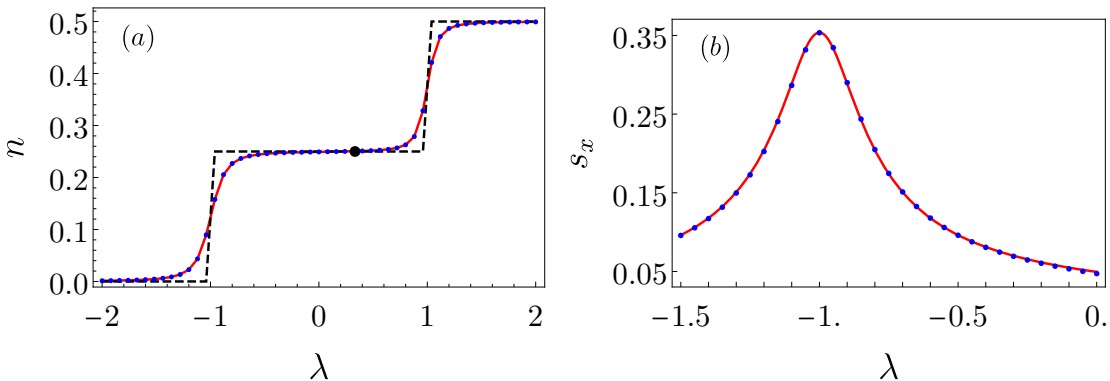

Figure 2: (a): Plot of the Rydberg excitation density $n$ as a function of $\lambda$ with $w = 0$ (black dotted line) and $w = 0.05$ (blue dots) for the two leg-ladder indicating the plateau at with $n = 1/4$ between $-1 \leq \lambda \leq 1$ as obtained using ED. The red solid line indicates $n$ as computed using variational wavefunction analysis around $\lambda = \pm 1$. The black circle at $\lambda \sim 0.33$ shows the position of the Ising transition. (b): Plot of $s_x$ as a function of $\lambda$ across the crossover at $\lambda \simeq -1$. The solid red line indicate results from the variational wavefunction approach while the blue dots indicates ED results corresponding to $N = 20$. All energies are scaled in units of $\Delta$. See text for details.

To show the efficacy of this approach, we compare results obtained via variational wavefunction method with exact numerics. To this end, we analyze $H_{\text{eff}}$ numerically using ED for $N \leq 40$ and using periodic boundary condition along the chains. We compute the expectation values $\hat{n}$ and

$$\hat{s}_x = \sum_{j=1}^{L}\sum_{\ell=1}^{2} \sigma_{2j,\ell}^x / N, \tag{11}$$

from both the variational wavefunction and using exact numerics. The former yields analytical expressions for $n$ and $s_x = \langle \hat{s}_x \rangle$ given by

$$n = \frac{1}{8}\sin^2\theta_0, \quad s_x = \frac{\sqrt{2}\sin 2\theta_0}{4}. \tag{12}$$

The plots of these as a function of $\lambda/\Delta$ with $w/\Delta = 0.05$ are compared with their exact numerical counterparts in Fig. 2(a) and 2(b). The plots show an excellent match; this shows that the crossover between phases with $n = 0$ to $n = 1/4$ is well captured by our variational analysis. A similar match is obtained for the crossover at $\lambda/\Delta \simeq 1$ as can be seen from Fig. 2(a); this has been presented in detail in App. A. We also note here that the extent of the plateau reduces with increasing $w$; we therefore work in the regime $w \ll \lambda, \Delta$ in this work.

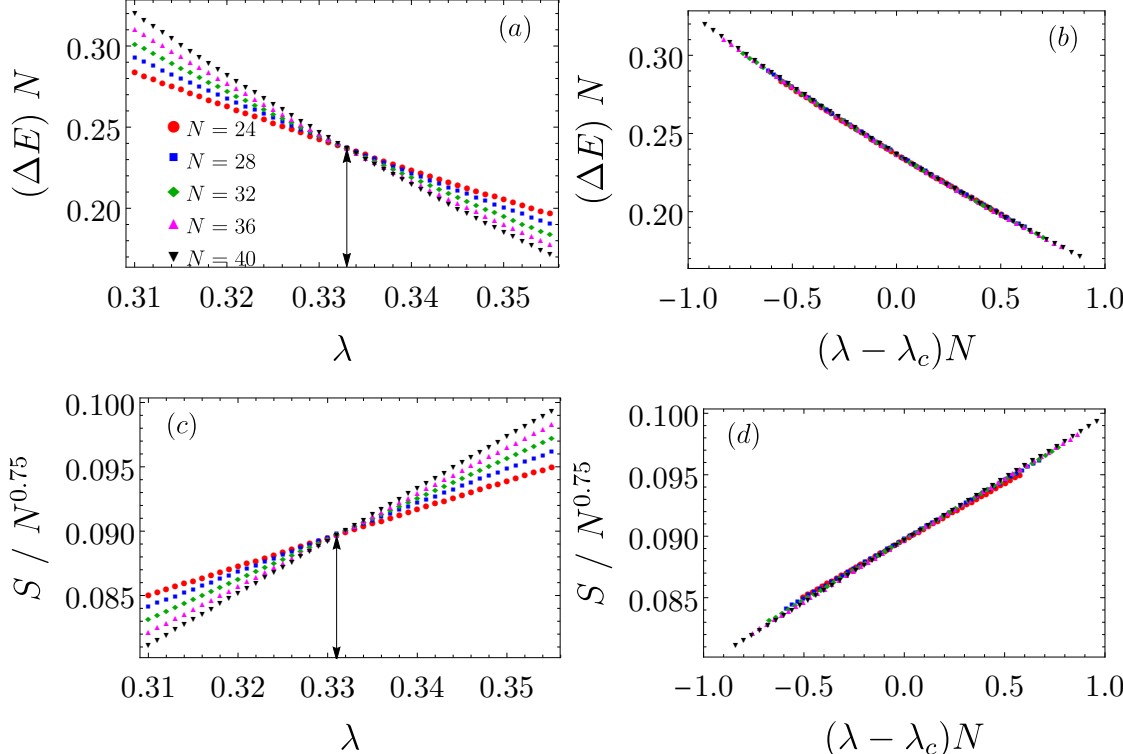

Figure 3: (a): Plot of $(\Delta E)N^z$ for a two-leg ladder as a function of $\lambda$ for $w = 0.05$ for several $N$ showing a crossing at $\lambda = \lambda_c \simeq 0.333$ for $z = 1$. (b) Plot of $(\Delta E)N^z$ as a function of $N^{1/\nu}(\lambda - \lambda_c)$ showing perfect scaling collapse for $z = \nu = 1$. (c): Plot of $SN^{2-z-\eta}$ as a function of $\lambda$ showing a crossing at $\lambda_c \simeq 0.331$ for $\eta = 0.25$. (d) Plot of $SN^{2-z-\eta}$ as a function of $(\lambda - \lambda_c)N^{1/\nu}$ showing perfect scaling collapse for $\nu = 1$ and $\eta = 0.25$. All energies are scaled in units of $\Delta$. See text for details.

Next, we concentrate on the nature of the ground state of the system for $-1 \le \lambda/\Delta \le 1$. Here the ground state is chosen by a quantum order-by-disorder mechanism from the manifold of $2^L$ classical Fock states with $n = 1/4$. Our numerical analysis reveals the presence of a quantum phase transition inside the plateau with $n = 1/4$ at $\lambda = \lambda_c$. We find numerically that the ground state of the system at $\lambda > \lambda_c$ hosts Rydberg excitations on even sites of *any one of the two chains*; it thus breaks a $\mathbb{Z}_2$ symmetry as shown schematically in Fig. 1(b). In contrast for $\lambda < \lambda_c$, the ground state is unique and constitute a superposition of all Fock states with $n = 1/4$ (Fig. 1(c)); it does not break any symmetry. This necessitates a transition at $\lambda = \lambda_c$

characterized by an order parameter

$$\hat{O} \;=\; \sum_{j=1}^{L}\sum_{\ell=1}^{2}(-1)^{\ell}\sigma_{2j,\ell}^{z}. \tag{13}$$

We note that the quantum phase transition is a result of the fact that the order-by disorder mechanism chooses two different ordered states, one of which breaks a discrete $\mathbb{Z}_2$ symmetry while the other does not.

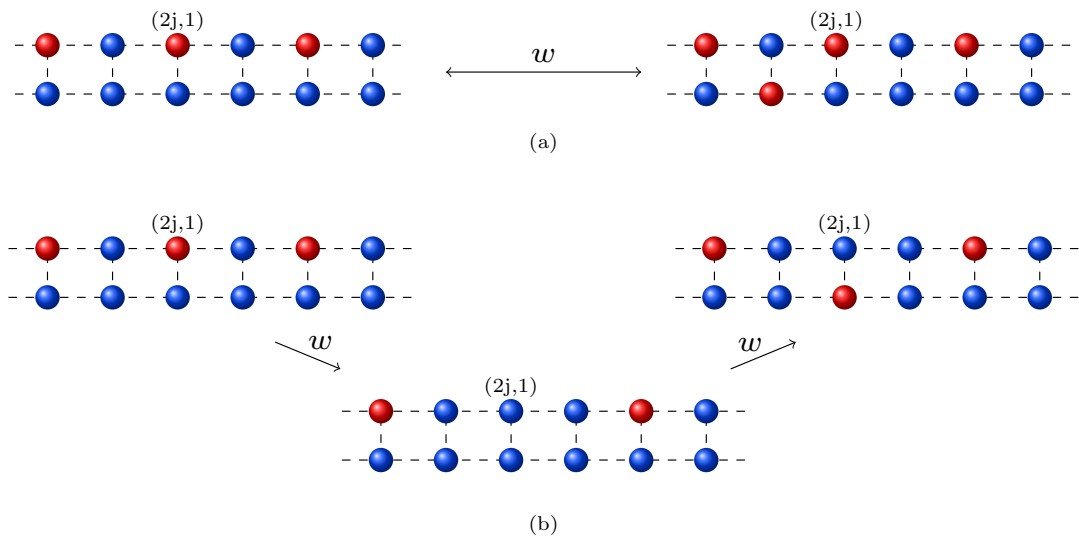

Figure 4: Schematic representation of the virtual processes leading to the effective Hamiltonian $H'$. The red (blue) circles represent atoms in Rydberg excited (ground) states. (a): The process which favors all Rydberg excitations to be on the even sites of one of the chains (chosen to be the top chain ($\ell = 1$) here) and leads to the first term in $H'$. The excited state constitutes one additional Rydberg excitation on an odd site of the bottom chain as shown schematically. (b) The competing virtual process which leads to the second term of $H'$. The excited state, which belongs to the high energy manifold, constitutes one additional Rydberg atom in its ground state at an even site of the top chain as shown. See text for details.

To check for the universality class of this transition, we carry out finite-size scaling analysis for $N \le 40$ on these ladders. We use the well-known scaling relations [17, 64]

$$\Delta E \;=\; N^{-z} f\left[N^{1/\nu}(\lambda-\lambda_c)\right], \tag{14}$$
$$S \;=\; \frac{1}{N}\langle \hat{O}^2\rangle = N^{2-z-\eta} g\left[N^{1/\nu}(\lambda-\lambda_c)\right],$$

where $z$ and $\nu$ are the dynamical critical and the correlation length exponents respectively, $\eta$ indicates the anomalous dimension of $\hat{O}$, $S$ is the order-parameter correlation, $f$ and $g$ are scaling functions, and $\Delta E$ denotes the energy gap between the ground and the first excited states. Such a scaling analysis, shown in Fig. 3 indicates the presence of a quantum phase transition at $\lambda_c \simeq 0.33\Delta$ with $z = \nu = 1$ and $\eta = 1/4$; these exponents confirm that the universality class is the same as the critical point of the classical 2D Ising model.

To obtain analytic insight into this transition, we construct a perturbative effective Hamiltonian using a Schrieffer-Wolff transformation as follows. First, we note that $w \ll |\lambda|, \Delta$ in the regime of interest. Using Eq. 6, we therefore write $H_{\mathrm{eff}} = H_0 + H_1$ where $H_1 = -w$

$\sum_{j,=1}^{2L} \sum_{\ell=1}^{2} \tilde{\sigma}_{j,\ell}^{x}$. We then use a canonical transformation operator $S'$ to write

$$H' = \exp[iS']H_{\text{eff}}\exp[-iS'] = H_0 + H_1 + [iS', H_0 + H_1] + \frac{1}{2}[iS', [iS', H_0]] + ..., \quad (15)$$

where the ellipsis indicates higher order terms. Next, following standard procedure, we eliminate all O($w$) terms in $H'$ which takes one out of the low-energy manifold of states. This allows to determine $S'$ using the resultant condition $[iS', H_0] = -H_1$ yielding

$$S' = w \sum_{j=1}^{2L} \sum_{\ell=1}^{2} (\Delta(-1)^j + \lambda)^{-1} \tilde{\sigma}_{j,\ell}^{y}. \quad (16)$$

Using the expression of $S'$, a straightforward calculation yields the effective Hamiltonian $H' = H_0 + [iS', [iS', H_0]]/2$. A subsequent projection to the manifold of states with $n = 1/4$ leads to [57, 65]

$$H' = \frac{-w^2}{\Delta - \lambda} \sum_{j=1}^{L} \Big[ \sum_{\ell=1}^{2} P_{2j,\ell} P_{2j+2,\ell} + \frac{\alpha_0}{2} \sum_{\ell,\ell'=1,2; \ell \neq \ell'} P_{2j,\ell'}(\sigma_{2j,\ell}^{x}\sigma_{2j,\ell'}^{x} + \sigma_{2j,\ell}^{y}\sigma_{2j,\ell'}^{y}) \Big], \quad (17)$$

where $\alpha_0 = (\Delta - \lambda)/(\Delta + \lambda)$. The virtual processes induced by $w$ which play a key role for realization of these two terms are schematically shown in Fig. 4.

The emergence of the phase transition at a critical value of $\lambda/\Delta$ can be understood by noting the competition between the two terms of $H'$ in Eq. 17. The first term, whose amplitude dominates for $\Delta - \lambda \simeq w$, prefers maximum possible Rydberg excitation on even sites of any one of the two chains; such a state is schematically shown in Fig. 1(b). The resultant ground state thus breaks $\mathbb{Z}_2$ symmetry. In contrast, the second term, which dominates when $\Delta + \lambda \simeq w$, prefers a linear superposition of all states with one Rydberg excitation on any chain of even sites of the ladders. The resultant ground state does not break any discrete symmetry. This necessitates an intermediate quantum critical point belonging to the 2D Ising universality class.

To estimate the position of this critical point, we start from the $\mathbb{Z}_2$ symmetry broken ground state with all Rydberg excitations on even sites of the first chain (Fig. 1(b)). The simplest excited state over these ground states constitutes equal superposition of Fock states with one Rydberg excitation on any one of two chains for a single even site $2j$; for $j' \neq 2j$, the Rydberg excitations occur on the first chain. This excited state can be represented as

$$|\psi_{\text{ex}}\rangle = \frac{1}{\sqrt{2}} \Big( |\downarrow_{1,1}\downarrow_{1,2}; \uparrow_{2,1}\downarrow_{2,2} .... \uparrow_{2j,1}\downarrow_{2j,2} ...; \uparrow_{2L,1}\downarrow_{2L,2}\rangle$$

$$+ |\downarrow_{1,1}\downarrow_{1,2}; \uparrow_{2,1}\downarrow_{2,2} .... \downarrow_{2j,1}\uparrow_{2j,2} ...; \uparrow_{2L,1}\downarrow_{2L,2}\rangle \Big), \quad (18)$$

where $|\psi_G\rangle = |\downarrow_{1,1}\downarrow_{1,2}; \uparrow_{2,1}\downarrow_{2,2} .... \uparrow_{2j,1}\downarrow_{2j,2} ...; \uparrow_{2L,1}\downarrow_{2L,2}\rangle$ is the symmetry broken ground state.

The energy cost of creating such an excited state can be easily computed as $\delta E_{\text{ex}} = \langle\psi_{\text{ex}}|H'|\psi_{\text{ex}}\rangle - \langle\psi_G|H'|\psi_G\rangle$. A straightforward calculation yields $\delta E_{\text{ex}} = \delta E_1 + \delta E_2$ where

$$\delta E_1 = \frac{w^2}{2(\Delta - \lambda)}, \quad \delta E_2 = -\frac{w^2}{\Delta + \lambda}. \quad (19)$$

We note that $\delta E_1$ originates from the first term of $H'$ which prefers all the Rydberg excitations to be on the same chain. In contrast, $\delta E_2$ comes from the second term which maximizes superposition of up-spins on different sites of the even rungs. An approximate estimate of the critical point can be obtained by equating these two energies; this yields the value of $\lambda$ at

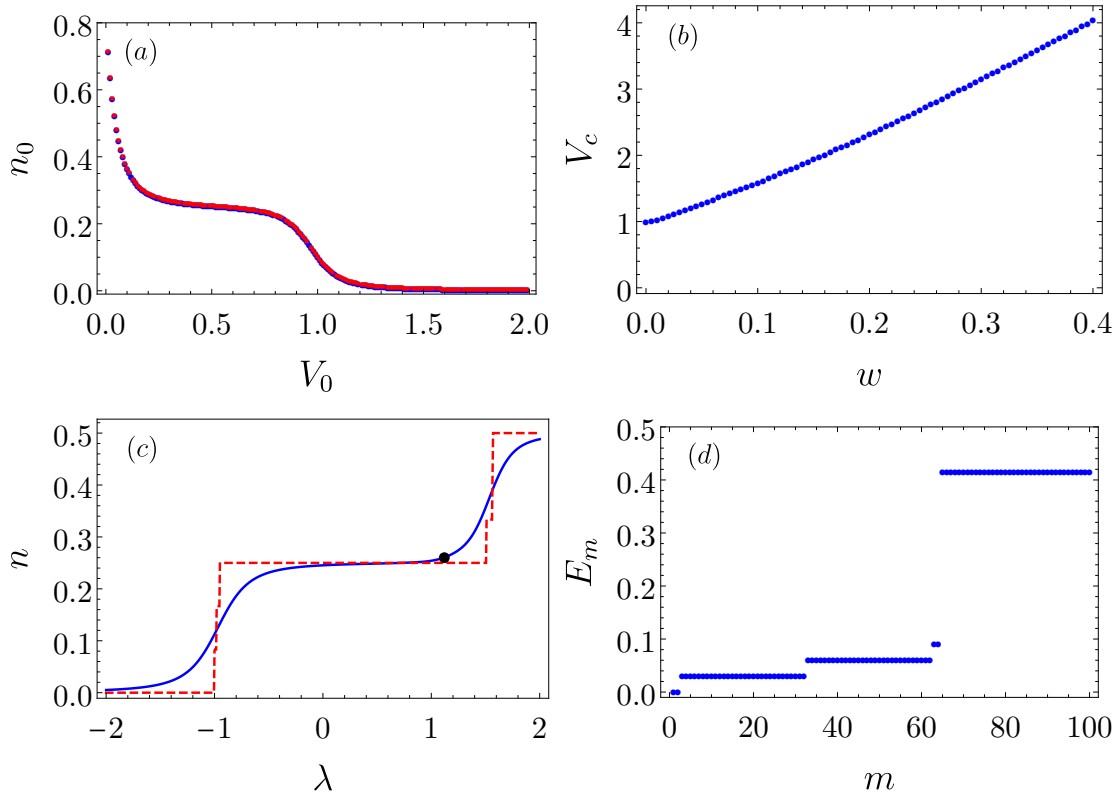

Figure 5: (a): Plot of $n_0$ as a function of $V_0$ for $w = 0.1$ and $\lambda = 1$ showing $V_c \simeq 1.6$ for $N = 12$ (blue squares) and $N = 16$ (red circles). (b) Plot of $V_c$ as a function of $w$ for $\lambda = 1$. (c) Plot of $n$ as a function of $\lambda$ for $w = 0.1$ and $V_0 = 2$ showing the plateau for $n$ and the position of the transition (black circle).(d) The eigenenergies $E_m$ of $H_{\text{eff}}^V$, measured from the ground state energy, as a function of $m$ (for $m \le 100$) for $\lambda = 1$, $w = 0$ and $N = 24$ showing the near-degenerate manifold. The two degenerate ground states at $w = 0$ correspond to two Fock states where the Rydberg excitations occur alternately on even sites of ladders 1 and 2. All energies are scaled in units of $\Delta$. See text for details.

which $\delta E_{\text{ex}} = 0$ so that the $\mathbb{Z}_2$ symmetry broken ground state becomes unstable to low-energy excitations. Such an estimate yields

$$\lambda_c = \Delta/3. \tag{20}$$

We note that this simple estimate matches remarkably well with that obtained from ED using finite-sized scaling analysis of the energy gap: $\lambda_c^{\text{ex}}/\Delta = 0.332$. This provides support for the fact that the transition is indeed realized due to an order-by-disorder mechanism which can be captured by the perturbative effective Hamiltonian $H'$ (Eq. 17).

We also note that the position of the transition can also be obtained as follows. First, we define pseudospin states on even sites of the ladder as $|1_{2j}\rangle \equiv |\uparrow_{2j,1}; \downarrow_{2j,2}\rangle$ and $|-1_{2j}\rangle \equiv |\downarrow_{2j,1}; \uparrow_{2j,2}\rangle$. In terms of Pauli matrices, $\vec{s}_j$, which acts on the space of these pseudospins, one can then identify $P_{2j,1(2)} = (1 - (+)s_{2j}^z)/2$. Using this identification, the first term of $H_{\text{eff}}$ (Eq. 37) can be written, ignoring an irrelevant constant term, as

$$\sum_{j=1}^{L}\sum_{\ell=1}^{2} P_{2j,\ell}P_{2j+2,\ell} \quad \rightarrow \quad \frac{1}{2}\sum_{j=1}^{L} s_{2j}^z s_{2j+2}^z \tag{21}$$

We then note that the second term of $H_{\text{eff}}$ (Eq. 37), acting on these pseudospin state, yields $|1_{2j}\rangle \Leftrightarrow |-1_{2j}\rangle$. This is due to the fact that the spins even sites of the ladder can not be both spin-up or both spin-down; the first condition follows from the constraint, while the second is a consequence of the ground state belonging to $n = 1/4$ sector. Using this, one can show

$$\frac{1}{2} \sum_{j=1}^{L} \sum_{\ell,\ell'=1,2;\ell\neq\ell'} P_{2j,\ell'}(\sigma_{2j,\ell}^{x}\sigma_{2j,\ell'}^{x} + \sigma_{2j,\ell}^{y}\sigma_{2j,\ell'}^{y}) \quad \rightarrow \quad \sum_{j=1}^{L} s_{2j}^{x} \tag{22}$$

Using Eqs. 21 and 22, we find an effective Ising representation of $H_{\text{eff}}$ given by

$$H_{\text{eff}}^{I} \quad = \quad \frac{-w^2}{\Delta - \lambda} \sum_{j=1}^{L} \left( \frac{1}{2} s_{2j}^{z} s_{2j+2}^{z} + \alpha_0 s_{2j}^{x} \right) \tag{23}$$

From this, one can, using Kramers-Wannier duality, read off the position of the critical point to be at $\alpha_0 = 1/2$ which also leads to $\lambda_c = \Delta/3$.

Before ending this section, we would like to point out that a similar model ladders with $\ell_0 > 2$ would lead to spin models which hosts $\mathbb{Z}_{\ell_0}$ symmetry broken ground states leading to realization of non-Ising quantum critical points. We show in the appendix, by an analysis similar to that carried out above, that such transition belong to 2D three-state Potts universality class for $\ell_0 = 3$. However, realization of $\ell_0 > 2$-leg ladders using Rydberg atoms chains with vdW interactions whose low-energy behavior is governed by $H_{\text{eff}}$ may be difficult; we discuss this further in the appendix.

## 4    Effect of vdW interaction

In this section, we shall discuss the fate of the Ising transition for a realistic vdW interaction given by Eq. 3. To this end, we concentrate in the regime where $V_0$ is large enough so that there is at most one Rydberg excitation in any pair of neighboring sites. To estimate the minimal $V_0 \equiv V_c$ required to satisfy this criteria, we diagonalize $H$ (Eq. 1) for a two-leg ladder with $N \leq 16$ and Hilbert space dimension $\mathcal{D} \leq 2^{16}$ to obtain its ground state $|\Psi_G\rangle$. We then compute

$$n_0 \quad = \quad \frac{1}{2} \sum_{\mathbf{r}\mathbf{r}'} \langle \Psi_G | n_{\mathbf{r}} n_{\mathbf{r}'} \delta_{|\mathbf{r}-\mathbf{r}'|-1} | \Psi_G \rangle \tag{24}$$

as a function of $V_0$ for a fixed $w$, $\lambda$, and $\Delta$. This allows us to estimate $V_c(w/\Delta, \lambda/\Delta)$ as the minimum value of $V_0$ for which $n_0 \leq \epsilon_0 \simeq 0.001$. This is shown in Fig. 5(a) where $n_0$ is plotted as a function of $V$ for $\lambda/\Delta = 1$ and $w/\Delta = 0.1$ for $N = 12, 16$. We note that $V_c$ is independent of $N$; thus one can safely use $V_0 > V_c$ to access the constrained Hilbert space for larger system sizes. A plot of $V_c/\Delta$ obtained from this procedure for $N = 16$ as a function of $w/\Delta$ for $\lambda/\Delta = 1$ is shown in Fig. 5(b). We find that $V_c$ increases with increasing $w$; this can be easily understood from the fact that increasing $w$ makes the freezing of spin on any site energetically more costly.

Next, we fix $V_0/\Delta = 2$ and $w/\Delta = 0.1$ so that the ground state does not have more than one Rydberg excitation on neighboring sites. We then diagonalize $H_{\text{eff}}^{V}$ (Eq. 5) for $N = 40$, obtain the ground state, and compute $n$ as a function of $\lambda/\Delta$ for $w = 0$ and $0.1\Delta$. Our results, shown in Fig. 5(c), indicate the presence of a wide range of $\lambda/\Delta$ with $n = 1/4$; for $w/\Delta = 0.1$, $n$ starts deviating from $1/4$ around $\lambda/\Delta \simeq 1.1$ and reaches its maximal value $1/2$ around $\lambda_u/\Delta \sim 2$. Our numerical, finite-size scaling analysis, also finds a Ising critical point around $\lambda = \lambda_c$ such that $\lambda_c/\Delta \simeq 1.12$ at the edge of the plateau. For $\lambda_c < \lambda < \lambda_u$, the system has a double degenerate ground state; it chose a single configuration only via spontaneous symmetry

breaking. This shows that the presence a finite second and higher neighbor interaction does not necessarily obliterate the Ising transition; however the transition is shifted to a higher value of $\lambda/\Delta$. We note that this is possible since the residual interaction term in $H_{\text{eff}}^V$ does not lift the degeneracy between the two $\mathbb{Z}_2$ symmetry broken ground states. Importantly, the presence of $V$ leads to a near-degenerate low-energy manifold with O($L$) states having $n \sim 1/4$; this is to be contrasted with the exact degenerate ground state manifold of $2^L$ states when the next nearest-neighbor interactions are neglected. This is shown in Fig. 5(d) for $\lambda/\Delta = 1.1$ and $w = 0$. The ground states for $w = 0$ shown in Fig. 5(d) correspond to two Fock states where Rydberg excitations occur alternatively on even sites of ladders 1 and 2. We also note that the presence of a finite $w$, whose magnitude is comparable to the width of the nearly-degenerate low-energy manifold, leads to a different ground state which constitutes a significant admixture of states within this manifold. The plot also indicates that one needs a finite but small $w$ so that states with $n \neq 1/4$ (higher excited states in Fig. 5(d)) are not strongly mixed with near degenerate O($L$) manifold of low-lying states.

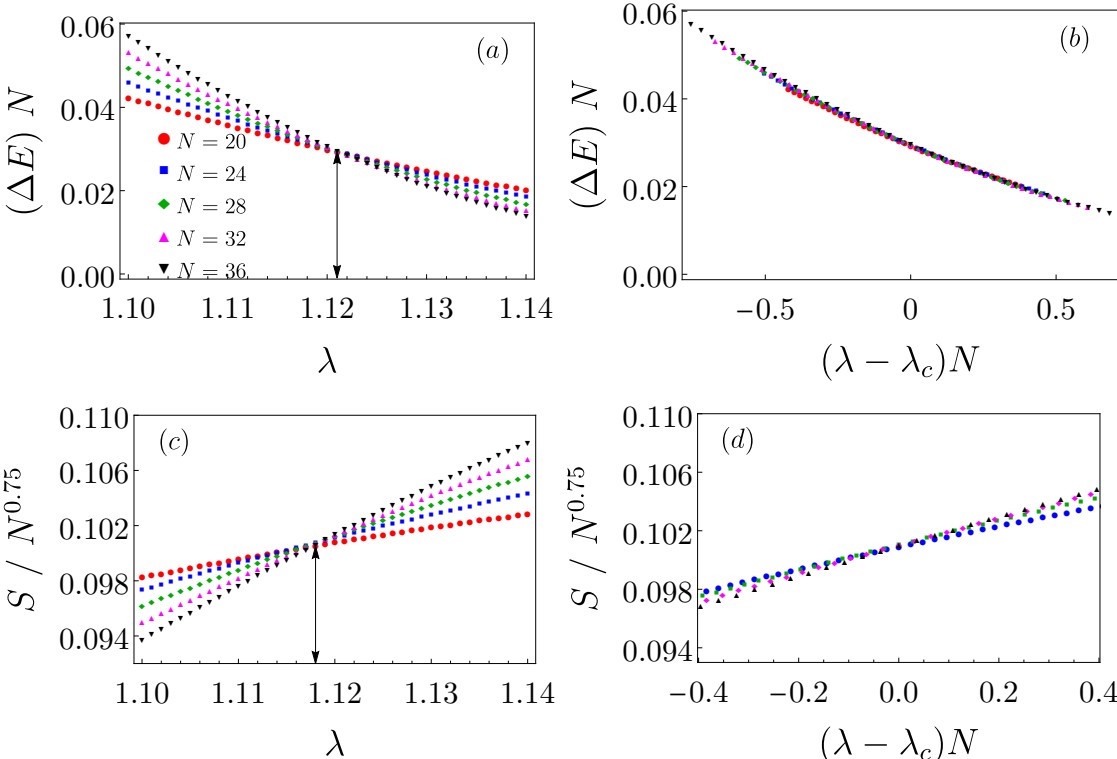

Figure 6: (a): Exact numerics confirming the presence of the Ising transition for $V_0 = 2.0$ and $w = 0.1$ (a): Plot of $(\Delta E)N^z$ for a two-leg ladder as a function of $\lambda$ for several $N$ showing a crossing at $\lambda = \lambda_c \simeq 1.121$ for $z = 1$. (b) Plot of $(\Delta E)N^z$ as a function of $N^{1/\nu}(\lambda - \lambda_c)$ showing perfect scaling collapse for $z = \nu = 1$. (c): Plot of $SN^{2-z-\eta}$ as a function of $\lambda$ showing a crossing at $\lambda_c \simeq 1.118$ for $\eta = 0.25$. (d) Plot of $SN^{2-z-\eta}$ as a function of $(\lambda - \lambda_c)N^{1/\nu}$ showing scaling collapse for $\nu = 1$, $\lambda_c \simeq 1.118$, and $\eta = 0.25$. All energies are scaled in units of $\Delta$. See text for details.

Next, we carry out finite-size scaling analysis near the transition. The results are indicated in Fig. 6. Our analysis reveals the presence of a quantum critical point at $\lambda_c \simeq 1.12\Delta$ for $V_0 = 2\Delta$ and $w = 0.1\Delta$. The Ising universality class of this transition is confirmed from finite-size scaling as shown in Fig. 6; our results indicate $z = \nu = 1$ and are consistent with $\eta = 0.25$ even though the scaling collapse of $S$ suggests significant finite-size scaling corrections for the

system sizes accessed using ED. The variation of $\lambda_c$ with $V_0 > V_c$ and $w$ is shown in Fig. 7(a). We note that the transition exists for a wide range of $V_0$ and $w$; $\lambda_c$ varies linearly with both $V_0$ and $w$ within this range. Moreover, we find that in contrast to its counterpart in $H_{\text{eff}}$, $\lambda_c$ depends on $w$ as shown in the inset of Fig. 7(a).

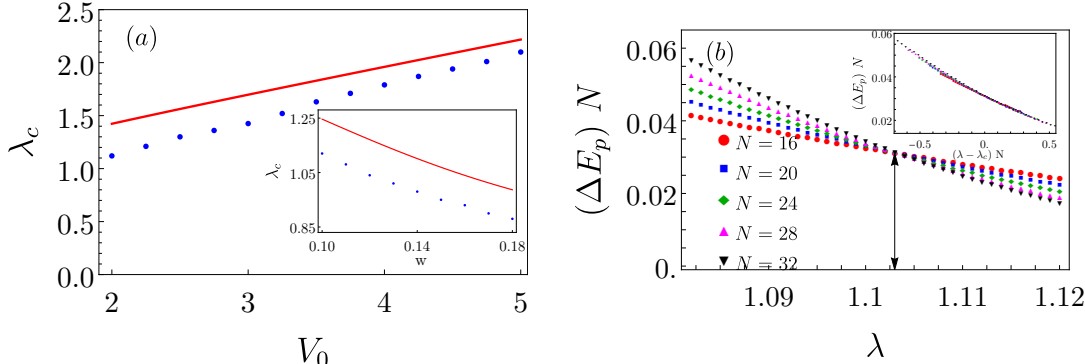

Figure 7: (a): Plot of $\lambda_c$ as a function of $V_0$ for $w_0 = 0.1$ showing the presence of the transition for a wide range of $V_0 > V_c$. The inset shows a plot of $\lambda_c$ as a function of $w$ for $V_0 = 2$. For both plots, the blue dots represent exact results obtained using ED while the red lines shows the analytical result obtained using Eq. 29. (b) A plot of $(\Delta E_p)N^z$ as a function of $\lambda$ for different $N$ showing a crossing at $\lambda_c \simeq 1.104$ for $z = 1$. The inset shows a plot $(\Delta E_p)N^z$ as a function of $(\lambda - \lambda_c)N^{1/\nu}$ showing data collapse for $z = \nu = 1$. These plots indicate that the Van-Vleck perturbation theory captures the essence of the Ising transition. All energies are scaled in units of $\Delta$. See text for details.

To obtain an analytic estimate of the shift in the position of the transition, we use a Van-Vleck perturbation theory to construct an effective low-energy Hamiltonian. To this end, we write $H_{\text{eff}}^V = H_0^V + H_1$ where $H_1 = -w \sum_{j=1}^{2L} \sum_{\ell=1}^{2} \tilde{\sigma}_{j,\ell}^x$ and $H_0^V$ can be read off from Eq. 5. We first identify the low-energy manifold of states corresponding to $H_0^V$; these states are represented as $|m\rangle$. Next, we construct a canonical transformation $S'$ leading to an effective Hamiltonian

$$H_V' = e^{iS'} H_{\text{eff}}^V e^{-iS'} = H_0^V + H_1 + [iS', H_0 + H_1] + \frac{1}{2}[iS', [iS', H_0^V]] + ..,  \quad (25)$$

where the ellipsis indicate higher order terms in $w$ which we neglect. Note that the presence of the interaction term in $H_0^V$ makes it difficult to obtain a closed form operator expression for $S'$; however, its matrix elements can still be analytically expressed. To determine $S'$, we again demand that it is chosen such that all O($w$) terms, which takes one out the low-energy manifold of states $|m\rangle$, are eliminated; this leads to matrix elements of $S'$ as

$$\langle n|iS'|m\rangle = \frac{\langle n|H_1|m\rangle}{E_n^0 - E_m^0}, \quad \langle m|iS'|m'\rangle = 0, \quad (26)$$

where $|n\rangle$ denotes eigenstates of $H_0^V$ which are not part of the low-energy manifold and $E_n^0$ are the corresponding eigenvalues. We also note that the second relation in Eq. 26 is automatically satisfied when $w$ is the large compared to energy width of the low-energy manifold (see Fig. 5(d)); this is the regime which we shall be interested in for the rest of this section. One can then substitutes Eq. 26 in Eq. 25 and obtain the matrix elements of $H_V'$ within the low-energy

manifold

$$
\begin{aligned}
\langle m|H_V'|m'\rangle &= E_m^0 \delta_{mm'} - \frac{w^2}{2} \sum_n \langle m| \sum_{j=1}^{2L} \sum_{\ell=1}^{2} \tilde{\sigma}_{j,\ell}^x |n\rangle \langle n| \sum_{j=1}^{2L} \sum_{\ell=1}^{2} \tilde{\sigma}_{j,\ell}^x |m'\rangle \\
&\times \left( \frac{1}{E_n^0 - E_m^0} + \frac{1}{E_n^0 - E_{m'}^0} \right),
\end{aligned}
\tag{27}
$$

where the sum over $n$ indicates sum over states outside the low-energy manifold. Here $E_m^0$ denotes the energy of the state $|m\rangle$ due to $H_0^V$ while the terms $\sim w^2$ results from second-order virtual processes due to $H_1$. We identify the low-energy manifold of states numerically and diagonalize $\langle m|H_V'|m'\rangle$ for several $N$. The resultant energy gap between the ground and the first excited state is denoted by $\Delta E_p$; a plot of $(\Delta E_p)N^z$ as a function of $\lambda$ is shown in Fig. 7(b) for $z = 1$. We find that it indicates the presence of quantum phase transition belonging to Ising universality at $\lambda_c \simeq 1.104\Delta$; the inset of Fig. 7(b) shows the plot of $(\Delta E_p)N^z$ as a function of $(\lambda - \lambda_c)N^{1/\nu}$ showing scaling collapse for $\lambda_c \simeq 1.104$ for $z = \nu = 1$. These results agree remarkably well with ED and shows that the essence of the Ising transition is well captured by the perturbation theory. We note that when $V_{\mathbf{rr'}}$ for $|\mathbf{r} - \mathbf{r'}| > 1$ are neglected, $E_m^0$ becomes a $m$ independent constant leading to an exactly degenerate manifold with all states having $n = 1/4$. The presence of a finite $V_{\mathbf{rr'}}$ for $|\mathbf{r} - \mathbf{r'}| > 1$ lifts this degeneracy and also allows the ground state manifold near the transition to have a small admixture of states with $n > 1/4$ in the presence of finite $w$ near the transition.

The picture of the transition that emerges from the above analysis is as follows. We note that Fig. 5(d) shows clear indication of near-degenerate low-energy manifold of states. The presence of a finite $w$ leads to a ground state which is an admixture of these low-energy nearly-degenerate manifold of states arising out of a quantum order-by-disorder mechanism. This leads to a unique ground state below a critical $\lambda = \lambda_c$; in contrast, for $\lambda > \lambda_c$, the ground state breaks a $\mathbb{Z}_2$ symmetry which necessitates the presence of an Ising critical point at $\lambda_c$.

To obtain an analytical estimate of the position of the critical point, we now approximate the ordered ground state for $\lambda > \lambda_c$ to be given by $|\psi_G\rangle$. This is not strictly correct since numerically we find that the ordered state has a small admixture of states with higher $n$. However, since this admixture is small, the estimate we obtain using this approximation is expected to be qualitatively correct. We then create an excited state given by $|\psi_{\text{ex}}\rangle$ (Eq. 18). Since both $|\psi_G\rangle$ and $|\psi_{\text{ex}}\rangle$ are part of low-energy manifold of states, we use Eq. 27 to compute $\langle H \rangle$ for them. A straightforward computation, keeping terms up to fourth nearest-neighbors in interaction ($|\mathbf{r} - \mathbf{r'}| \leq \sqrt{5}$), shows that

$$
\begin{aligned}
E_{\text{ex}}^V &= \langle \psi_{\text{ex}}|H_v'|\psi_{\text{ex}}\rangle = E_G^V + \frac{\delta E_1^V}{2} + \delta E_2^V, \quad E_G^V = \langle \psi_G|H_v'|\psi_G\rangle, \\
E_G^v &= L\left( -(\Delta + \lambda) + \frac{V_0}{2^6} - \frac{w^2}{(\Delta - \lambda) + V_0/4} \right) + E_0 \\
\delta E_1^V &\simeq 2V_0 n_1 + \frac{w^2}{\frac{V_0}{4} + (\Delta - \lambda)}, \quad \delta E_2^V \simeq -\frac{w^2}{\Delta + \lambda}, \quad n_1 = \left( \frac{1}{(\sqrt{5})^6} - \frac{1}{2^6} \right),
\end{aligned}
\tag{28}
$$

where $E_0$ is the energy of the all-spin down state and we have neglected terms $O(w^2 V_0 /(2^6(\Delta + \lambda)^2))$ and $O(w^2 V_0/((\sqrt{5})^6(\Delta + \lambda)^2))$; these terms are always small in the range of $V_0$ and $w$ we are interested in. The virtual processes responsible for the $O(w^2)$ term turns out to be same that in Fig. 4; their amplitudes, however, change due to the presence of $V_0$. We note that for $w = 0$, $\delta E_{\text{ex}} < 0$ which means the ordered phase never occurs in absence of the order-by-disorder mechanism realized via $O(w^2)$ terms in $H_V'$. Second, for $V_0 = 0$, $\delta E_{1(2)}^V = \delta E_{1(2)}$

and we get back Eq. 19 for the excitation energy. Equating $\delta E_{\text{ex}}^V = 0$, we finally find

$$\lambda_c = \frac{V_0^2 n_1 + 6w^2 - \sqrt{(V_0 n_1 (8\Delta + V_0) - 2w^2)^2 + 32w^4}}{8V_0 n_1}. \tag{29}$$

We note that $V_0 = 2\Delta$ and $w = 0.1\Delta$, Eq. 29 yields $\lambda_c \simeq 1.25\Delta$ which is close to the exact result obtained from ED: $\lambda_c^{\text{exact}} = 1.12\Delta$. A plot of $\lambda_c$ as a function of $V$ and $w$ shown in Fig. 7(a). Comparting the analytical result (red solid line) with the numerical ones (blue dots), we find that the former overestimates $\lambda_c$ by a small amount but provides similar dependence of $\lambda_c$ on both $V$ and $w$. The numerical difference between the two arises from the perturbative nature of the Van-Vleck result and also from the approximate nature of the ordered ground state since we have neglected the small admixture with states having $n > 1/4$. We note that the presence of a critical lower value of $w$ is clearly required for $\lambda_c < \lambda_u$; for $w < w_c$, which can be estimated by equating $\lambda_c = \lambda_u$ in Eq. 29, the transition does not occur.

Thus our analysis shows that the presence of a finite $V_0$ does not obliterate the Ising transition. This allows for the possibility of concrete realization of such a transition in experimentally relevant Rydberg atom systems. Importantly, the perturbative analysis clearly indicates that the transition is stabilized by a quantum order-by-disorder mechanism.

## 5  Discussion

In this work we have identified a quantum phase transition which is stabilized by a quantum order-by-disorder mechanism in Rydberg ladders with staggered detuning. Such a transition does not have any analogue in Rydberg chains studied earlier. We have shown that this transition persists for a wide range of vdW interaction strength between the Rydberg atoms. Our numerical studies are supplemented by perturbative calculations which provides analytical insight into the nature of the transition via identification of the competing terms in the low-energy effective Hamiltonian of the system responsible for it; moreover, they provide remarkably accurate estimate of the critical detuning.

Our prediction can be tested using standard experiments on Rydberg atom systems [25, 26]. The simplest geometry to consider would be the two-leg ladder with $w \ll |\lambda|, \Delta$ and a finite $V_0$ within range shown in Fig. 7(a). We envisage a situation where the detunings at the odd(even) sites of the Rydberg chains, given by $\lambda_{\text{odd}}(\lambda_{\text{even}})$, are

$$\lambda_{\text{even}} = \lambda + \Delta, \quad \lambda_{\text{odd}} = \lambda - \Delta. \tag{30}$$

The Ising transition is then expected to occur, for example, for $V_0 = 3\Delta$, at $\lambda_c \simeq 1.5\Delta$ which translates to $\lambda_{\text{even}}/\lambda_{\text{odd}} \simeq 5$. These values can of course be tuned by appropriate tuning of $V_0$ and $w$ within the specified range discussed in Sec. 4. In the ordered phase, with $\lambda > \lambda_c$, all the Rydberg excitations will preferentially happen on one of the two ladder provided one breaks the $\mathbb{Z}_2$ symmetry by changing the detuning slightly on any even site of the one of the chains. In contrast, the Rydberg excitations in the same setting will be distributed on both chains for $\lambda_{\text{even}}/\lambda_{\text{odd}} < 5$. This change should be easily detected by standard fluorescent imaging techniques used for detection of Rydberg excitations in these systems [25–28].

Finally, our analysis of the Rydberg ladder in the limit $V_{\mathbf{rr}'} \ll w, \lambda, \Delta$ points towards an interesting class of spin models where $\mathbb{Z}_{\ell_0}$ symmetry broken ground states and associated quantum phase transitions are stabilized by an order-by-disorder mechanisms. For a ladder with $\ell_0$ chains, as shown in App. B, a $\mathbb{Z}_{\ell_0}$ symmetry broken ground state is realized for $\lambda > \lambda_c = \Delta(\ell_0 - 1)/(\ell_0 + 1)$; in contrast, for $\lambda < \lambda_c$, the ground state does not break any symmetry. This leads to the realization of a critical point at $\lambda = \lambda_c$ which, for example, belongs to three-state Potts university class for $\ell_0 = 3$. However, unlike $\ell_0 = 2$, ladders with

$\ell_0 > 2$ chains also require periodic boundary conditions along the rung direction to stabilize non-Ising transitions; this might be difficult to achieve in current experimental setups.

In conclusion we have studied the phase diagram of Rydberg ladders with $\ell_0$ legs in the presence of staggered detuning. Our study indicates that the low-energy behavior of such systems is described by class of constrained spin models which support $\mathbb{Z}_{\ell_0}$ symmetry broken ground states and associated emergent quantum criticality stabilized by an order-by-disorder mechanism. For the two-leg ladder, we show, by considering realistic vdW interactions, that the presence of such an interaction do not obliterate the emergent Ising transition; this leads to the possibility of its detection in standard experimental setup.

## Acknowledgement

MS thanks Roopayan Ghosh for discussions. KS thanks DST, India for support through SERB project JCB/2021/000030.

## A  Variational wavefunction approach for $\lambda \simeq \Delta$

In this section, we shall address the ground state of the system around $\lambda = \Delta$ and for $\lambda, \Delta \gg w$. To this end, we first note, that for $\lambda \gg \Delta, w$, the ground state of the system prefers maximal number of Rydberg excitations and can therefore be written as

$$
\begin{aligned}
|\psi_3\rangle &= \cos\phi_1|\psi_{3a}\rangle + \sin\phi_1|\psi_{3b}\rangle \\
|\psi_{3a}\rangle &= \prod_{j=1}^{L/2} |\uparrow_{2j-1,1}, \downarrow_{2j-1,2}; \downarrow_{2j,1}, \uparrow_{2j,2}\rangle, \quad |\psi_{3b}\rangle = |\downarrow_{2j-1,1}, \uparrow_{2j-1,2}; \uparrow_{2j,1}, \downarrow_{2j,2}\rangle, \quad (31)
\end{aligned}
$$

where once again we have chosen $\phi_1$ to be independent of $j$. In contrast, for $\lambda \leq \Delta$, the ground state prefers to have down-spins on the odd sublattices. Thus one may choose the variational ground state wavefunction to be $|\psi_4(\phi_2)\rangle \equiv |\psi_4\rangle = |\psi_2(\phi = \phi_2)\rangle$ where $|\psi_2\rangle$ is given by Eq. 8. Using these wavefunctions, one can therefore construct a variational wavefunction given by

$$
|\Phi_v\rangle = \cos\beta|\psi_3\rangle + \sin\beta|\psi_4\rangle, \quad (32)
$$

where $0 \leq \beta \leq \pi/2$. The corresponding variational energy can be computed in a straightforward manner and yields

$$
E_{1v} = \langle\Phi_v|H_2|\Phi_v\rangle = E_0 + (\lambda - \Delta)\cos^2\beta - w\sin(2\beta)\cos(\phi_1 - \phi_2). \quad (33)
$$

The minimization of $E_{1v}$ leads to

$$
\phi_1 = \phi_2, \quad \beta_0 = \frac{1}{2}\arctan\left[\frac{2w}{\Delta - \lambda}\right]. \quad (34)
$$

Next, we consider the change in expectation values of the operators $\hat{n}$ and $s_x$ across the transition. To this end, we note that using Eqs. 32 and 34, we find

$$
\begin{aligned}
\langle\hat{n}\rangle_v &= \frac{1}{2} - \frac{1}{4}\cos^2\beta_0 = \frac{1}{2} - \frac{1}{8}\left(1 - \frac{\lambda - \Delta}{\sqrt{(\lambda - \Delta)^2 + 4w^2}}\right), \\
\langle s'_x\rangle_v &= \frac{1}{N}\langle\sum_{j=1}^{L}\sum_{\ell=1}^{2}\langle\sigma^x_{2j-1,\ell}\rangle = \frac{\sin 2\beta_0}{4} = \frac{w}{2\sqrt{(\lambda - \Delta)^2 + 4w^2}}.
\end{aligned} \quad (35)
$$

These expectations values, can also be computed using the exact ground state obtained from ED. The two results show excellent match similar to their counterparts near $\lambda \sim -\Delta$ discussed in the main text. The plot for $n$ near $\lambda \sim \Delta$, shown in Fig. 2(a), corroborates the above statement. We note here that the two states $|\psi_3\rangle$ and $|\psi_4\rangle$ belong to two different $\mathbb{Z}_2$ orders corresponding to $n = 1/4$ and $n = 1/2$ respectively. Thus it is natural to expect a first order transition between them around $\lambda \simeq \Delta$. However, our numerics from ED seems to indicate a crossover which can be inferred from $N$ independence of the data. This is most likely a consequence of the presence of other sector states with fixed $n$. These states do not feature in this simple minded variational wavefunction computation and their presence may change this transition to a crossover.

## B    $\ell_0$-leg ladder

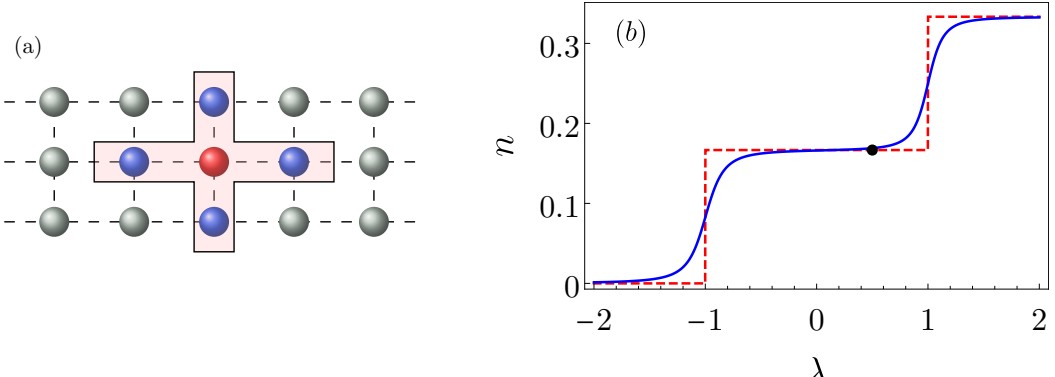

Figure 8: (a) Schematic representation of the Rydberg ladder for $\ell_0 = 3$ showing the Rydberg blockade radius (rectangular region) around a Rydberg excitation (red circle). The blue circles denote sites with blocked Rydberg excitation. The black circle at $\lambda \sim 0.5$ shows the position of the quantum phase transition. (b) The plot of Rydberg excitation density $n$ for a $\ell_0 = 3$ leg ladder as a function of $\lambda$. The red dotted line corresponds to $w = 0$ showing first order transition between the plateaus. The blue solid lines correspond to $w = 0.05$. The black circle shows the position of the quantum phase transition. All energies are scaled in units of $\Delta$. See text for details.

In this section, we consider the model $H_{\text{eff}}$ for $\ell_0$-legged ladder. For $\ell_0 > 2$, such ladders are difficult to realize in an experimentally relevant Rydberg atom arrays since periodic boundary conditions is required along the rung direction. However we study such model Hamiltonians here since they support phase transitions with non-Ising universality that are stabilized by order-by-disorder mechanism. In what follows, we shall provide a general expression for the low-energy effective Hamiltonian $H'_{\ell_0}$ for a ladder with $\ell_0$ legs. We shall also provide numerical evidence of the transition for $\ell_0 = 3$ legged ladder.

To this end, we consider an arrangement of $\ell_0$ Rydberg chains each having $2L$ sites so that $N = 2L\ell_0$ as shown in Fig. 8(a) for $\ell_0 = 3$. As shown in this figure, we consider a situation where the presence of a Rydberg excited atom on any site precludes the presence of another such excitation in all of the $\ell_0 - 1$ sites of the same rung of the ladder. The model Hamiltonian

that we shall consider for these chains is a generalization for $H_{\text{eff}}$ (Eq. 6) and is given by

$$
H = -\sum_{j=1}^{2L}\sum_{\ell=1}^{\ell_0}\left(w\tilde{\sigma}^x_{j,\ell}+\frac{1}{2}[\Delta(-1)^j+\lambda]\sigma^z_{j,\ell}\right), \quad \tilde{\sigma}^x_{j,\ell}=P_{j-1,\ell}\left(\prod_{\ell'\neq\ell}P_{j,\ell'}\right)\sigma^x_{j,\ell}P_{j+1,\ell}. \tag{36}
$$

The phase diagram for such chains for $w = 0$ can be obtained in a straightforward manner. For $\lambda < 0$, and $|\lambda| \gg \Delta$, $n = 0$. Around $|\lambda| \simeq \Delta$, it becomes energetically favorable to create one Rydberg excitation on one of the $\ell_0$ sites of an even rung. This leads to a classical ground state with $n_0 = 1/(2\ell_0)$; the ground state is macroscopically degenerate since there are $\ell_0^L$ Fock states with $n = 1/(2\ell_0)$. For $\lambda > \Delta$, the ground state corresponds to maximum possible Rydberg excitations leading to $n = 1/\ell_0$; it exhibits a $\ell_0$ fold degeneracy. For $w = 0$, there are first-order transitions between these ground states at $\lambda/\Delta \simeq \pm 1$; for finite $w$, these are related by crossovers. The representative phase diagram, for $\ell_0 = 3$, is shown in Fig. 8(b). Our numerical analysis also detects, for $\ell_0 = 3$, a quantum transition within the $n = 1/6$ plateau at $\lambda_c/\Delta \simeq 0.52$. A finite-sized scaling analysis near of the energy gap $\Delta E$, as shown in Fig. 9(a) and 9(b) indicates $z = 1$ and $\nu = 5/6 \simeq 0.83$ for the transition. Furthermore we define an order parameter $\hat{O}_{\ell_0} = \sum_{j=1}^L \sum_{\ell=1}^{\ell_0} \exp[2\pi i\ell/\ell_0]\sigma^z_{2j,\ell}$ and computes its correlation function for the $\ell_0 = 3$ chain: $S = \langle\hat{O}_3^*\hat{O}_3\rangle/N$. We find that, using a finite sized scaling analysis, $S \sim N^{2-z-\eta}$ with $\eta = 4/15 \simeq 0.267$ and $z = 1$ as shown in Figs. 9(c) and 9(d). Thus we conclude that the transition belongs to the 2D three-state Potts universality class.

An analytic insight into this transition is provided by a calculation similar to that outlined in the main text leading to Eq. 17. For an $\ell_0$-leg the effective Hamiltonian receives contribution from processes which are similar to those shown in Fig. 4 (generalized for $\ell_0$ legs) and is given by [57, 65]

$$
H^{\ell_0}_{\text{eff}} = \frac{-w^2}{\Delta-\lambda}\sum_{j=1}^L\Big[\sum_{\ell=1}^{\ell_0}P_{2j,\ell}P_{2j+2,\ell}+\frac{\alpha_0}{2}\sum_{\ell,\ell'=1,\ell_0;\ell\neq\ell'}P_{2j,\ell'}(\sigma^x_{2j,\ell}\sigma^x_{2j,\ell'}+\sigma^y_{2j,\ell}\sigma^y_{2j,\ell'})\Big],
$$
$$\tag{37}$$

where $\alpha_0 = (\Delta - \lambda)/(\Delta + \lambda)$. The emergence of the phase transition at a critical value of $\lambda/\Delta$ can once again be understood by noting the competition between the two terms in Eq. 37. The first term prefers a $\mathbb{Z}_{\ell_0}$ symmetry broken ground state with all Rydberg excitations on any one of the $\ell_0$ chains; in contrast, the second term prefers a linear superposition of all states with one Rydberg excitation on any site of even rungs of the ladder. This necessitates an intermediate quantum critical point belonging to $\mathbb{Z}_{\ell_0}$ universality class.

To estimate the position of this critical point, we once again consider an excited state over the $\mathbb{Z}_{\ell_0}$ symmetry broken ground state $|\psi'_G\rangle$ given by

$$
|\psi'_G\rangle = |\downarrow_{1,1}\downarrow_{1,2}..\downarrow_{1,\ell_0};\uparrow_{2,1}\downarrow_{2,2}..\downarrow_{2,\ell_0};....\uparrow_{2j,1}\downarrow_{2j,2}..\downarrow_{2j,\ell_0}...;\uparrow_{2L,1}\downarrow_{2L,2}..\downarrow_{2\ell_0}\rangle. \tag{38}
$$

Such an excited state can be created by constructing a linear superposition of states with a Rydberg excitation on any one of the sites of one of the even rungs of the ladder for which $j' = 2j$; for all $j' \neq 2j$, the Rydberg excitation reside at $\ell_0 = 1$. Such an excited state can be represented as

$$
\begin{aligned}
|\psi'_{\text{ex}}\rangle = \frac{1}{\sqrt{\ell_0}}\big(&|\downarrow_{1,1}\downarrow_{1,2}..\downarrow_{1,\ell_0};\uparrow_{2,1}\downarrow_{2,2}..\downarrow_{2,\ell_0};....\uparrow_{2j,1}\downarrow_{2j,2}..\downarrow_{2j,\ell_0}...;\uparrow_{2L,1}\downarrow_{2L,2}..\downarrow_{2\ell_0}\rangle \\
&+|\downarrow_{1,1}\downarrow_{1,2}..\downarrow_{1,\ell_0};\uparrow_{2,1}\downarrow_{2,2}..\downarrow_{2,\ell_0};....\downarrow_{2j,1}\uparrow_{2j,2}..\downarrow_{2j,\ell_0}...;\uparrow_{2L,1}\downarrow_{2L,2}..\downarrow_{2\ell_0}\rangle \\
&+....|\downarrow_{1,1}\downarrow_{1,2}..\downarrow_{1,\ell_0};\uparrow_{2,1}\downarrow_{2,2}..\downarrow_{2,\ell_0};....\downarrow_{2j,1}\downarrow_{2j,2}..\uparrow_{2j,\ell_0}...;\uparrow_{2L,1}\downarrow_{2L,2}..\downarrow_{2\ell_0}\rangle\big),
\end{aligned}
$$
$$\tag{39}$$

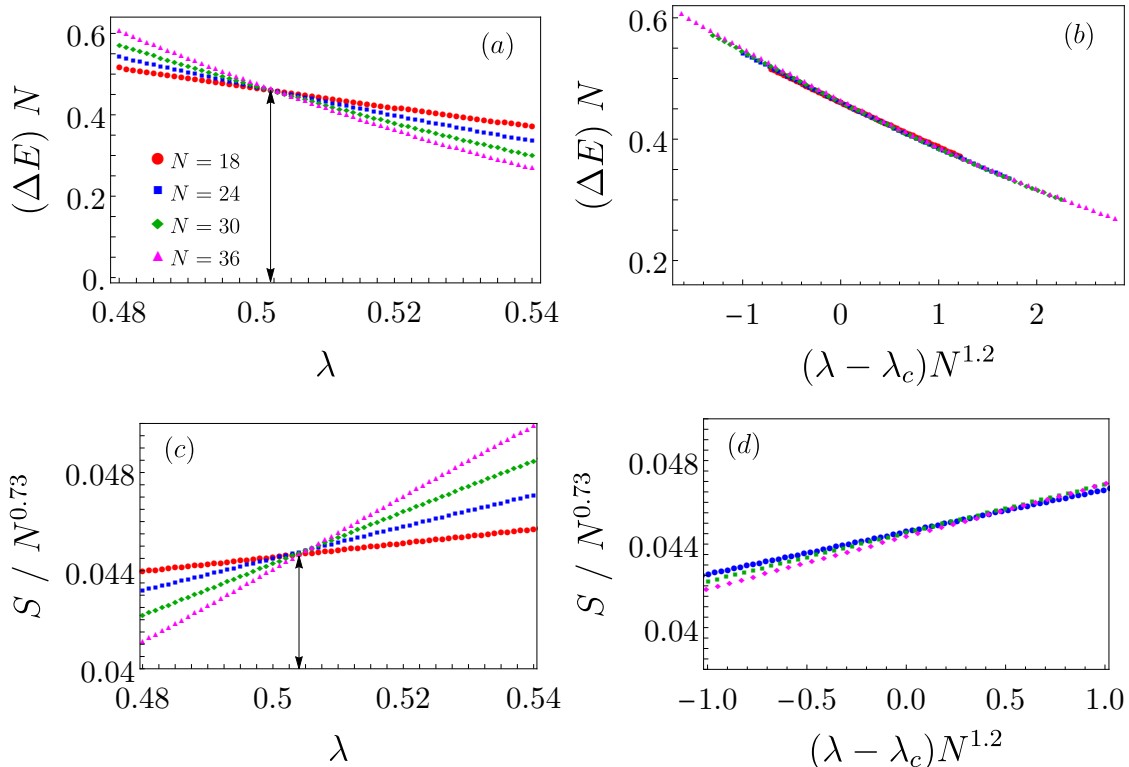

Figure 9: Exact numerics confirming the presence of the three-state Potts transition for $\ell_0 = 3$. (a): Plot of $(\Delta E)N^z$ for a three-leg ladder as a function of $\lambda$ for several $N$ showing a crossing at $\lambda = \lambda_c \simeq 0.501$ for $z = 1$. (b) Plot of $(\Delta E)N^z$ as a function of $N^{1/\nu}(\lambda - \lambda_c)$ showing perfect scaling collapse for $z = 1$ and $\nu \simeq 1/1.2 = 0.83$.(c): Plot of $SN^{2-z-\eta}$ as a function of $\lambda$ showing a crossing at $\lambda_c \simeq 0.502$ for $\eta \simeq 0.27$. (d) Plot of $SN^{2-z-\eta}$ as a function of $\lambda - \lambda_c)N^{1/\nu}$ showing a scaling collapse for $\nu = 1/1.2$ and $\eta = 0.27$. All energies are scaled in units of $\Delta$. See text for details.

The energy cost of creating such an excited state can be easily computed following the method outlined in the main text and leads to $\delta E_{\text{ex}}^{\ell_0} = \delta E_1^{\ell_0} + \delta E_2^{\ell_0}$ where

$$\delta E_1^{\ell_0} = \frac{w^2(1 - 1/\ell_0)}{\Delta - \lambda}, \quad \delta E_2^{\ell_0} = -\frac{w^2(\ell_0 - 1)}{\Delta + \lambda}. \tag{40}$$

The energy gap vanishes when $\delta E_{\text{ex}}^{\ell_0} = 0$ which leads to an estimate of the critical point given by

$$\lambda_c = \Delta(\ell_0 - 1)/(\ell_0 + 1). \tag{41}$$

For $\ell_0 = 2$, this reproduces Eq. 20 of the main text; for $\ell_0 = 3$, it provide an estimate of the three-state Potts critical point to be $\lambda_c = \Delta/2$ which matches quite well with exact numerical results shown Fig. 9.

Before ending this section, we note that we expect that for $\ell_0 > 4$, the transition shall be first order; however we have not been able to carry out ED studies on sufficiently large system size to ascertain this due to increased dimension of the constrained Hilbert space.

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
