# Peer review of "Quantum order-by-disorder induced phase transition in Rydberg ladders with staggered detuning"

_SciPost Physics_

## Round 1 · Referee Report · Anonymous (Referee 1) · 2022-6-30

Strengths

  1. Rydberg lattice are readily realisable in the state of the art experiments.
  2. An interesting phenomenon of order by disorder transition is discussed.

Weaknesses

The draft does not explicitly discuss how to realise experimentally the staggered detuning in a Rydberg chain.

Report

The authors studied the ground state configurations in a Rydberg ladder in the presence of both staggered (Delta) and uniform (lambda) detunings. In the case of the nearest neighbour blockade, the system exhibits macroscopic degenerate states with a Rydberg filling of n=1/4 for a finite range of lambda/Delta ratio. Within the n=1/4 plateau, above a critical value of uniform detuning, the system chooses one of the classical configurations in which Rydberg atoms are excited only in one of the chains, breaking the Z2 symmetry via quantum order by disorder mechanism. In Sec. 3, the physics is analyzed using effective Hamiltonians. In contrast, in Sec. 4, the original Hamiltonian for a Rydberg ladder is used and shows the existence of the critical point in the n=1/4 plateau. The quantum order by disorder transition emerges from the staggered detuning and the Rabi coupling. Finally, the study is extended beyond the two-leg ladder scheme. In particular, the three-leg ladder possesses a quantum critical point with a three-states Pott universality class in 2D. This work is exciting; at first, it was less obvious, but later it became more apparent in the draft. I recommend the paper to publish in the scipost physics. More specific comments are provided below.

Requested changes

Specific comments:

  1. On multiple occasions in the draft, calling “amplitude” of realizing the Rydberg excitation at any site as “detuning” is very misleading, especially from an atom-light interaction point of view. Detuning has units of frequency (equivalently energy) and well-defined interpretation even without any Rydberg excitations.

  2. One of the essential ingredients of the model is the staggered detuning; it would be beneficial if the authors could provide a scheme to implement such a pattern in detuning along the ladder. In any case, it requires local manipulations using other optical fields and may not be as straightforward as the authors think. Along these lines, the papers cited in the introduction, Refs [50-59] do not provide how to implement that either.

  3. Note that quantum order by disorder transition is also discussed using a Rydberg + square lattice setup in https://journals.aps.org/prx/pdf/10.1103/PhysRevX.4.041037 but in a completely different interaction scheme.

  4. In Eq. 1, the Hamiltonian for the Rydberg ladder, the Rabi term or the transverse magnetic field term (\sigma_x) has an additional negative sign, in contrast to the Hamiltonians describing the original Rydberg setups. It doesn’t look like the sign matters for the Physics discussed in the manuscript, as in the effective Hamiltonian in Eq. 17, it is w^2 in the first term, but it is better to clarify it.

  5. When the exact 1/r^6 interaction is considered, the transition point shifts to a larger value of \lambda, and a further increase in lambda, makes a jump in the Rydberg density to n=1/2. Then, mentioning lambda>lambda_c, the system chooses a single classical configuration in the presence of weak quantum fluctuations arising from the Rabi coupling. Is not better to say, lambda_c < lambda < lambda_u, where lambda_u defines the upper value of n=1/4 plateau, the system chooses a single configuration?

  6. If we do an ED with the model (1) and truncate to nearest neighbour interaction, do we get the critical lambda=0.33 Delta?

  7. It is also good to comment on what happens if the Rabi coupling becomes sufficiently large. The ground state will have contributions from the degenerate manifold of the first excited state in Fig. 5d.

  • validity: top
  • significance: top
  • originality: top
  • clarity: high
  • formatting: excellent
  • grammar: excellent

Author:  Krishnendu Sengupta  on 2022-08-09  [id 2717]

(in reply to Report 1 on 2022-06-30)

Referee: The authors studied the ground state configurations in a Rydberg ladder in the presence of both staggered (Delta) and uniform (lambda) detunings. In the case of the nearest neighbour blockade, the system exhibits macroscopic degenerate states with a Rydberg filling of n=1/4 for a finite range of lambda/Delta ratio. Within the n=1/4 plateau, above a critical value of uniform detuning, the system chooses one of the classical configurations in which Rydberg atoms are excited only in one of the chains, breaking the Z2 symmetry via quantum order by disorder mechanism. In Sec. 3, the physics is analyzed using effective Hamiltonians. In contrast, in Sec. 4, the original Hamiltonian for a Rydberg ladder is used and shows the existence of the critical point in the n=1/4 plateau. The quantum order by disorder transition emerges from the staggered detuning and the Rabi coupling. Finally, the study is extended beyond the two-leg ladder scheme. In particular, the three-leg ladder possesses a quantum critical point with a three-states Pott universality class in 2D. This work is exciting; at first, it was less obvious, but later it became more apparent in the draft. I recommend the paper to publish in the scipost physics. More specific comments are provided below.

Response: We thank the referee for a careful reading of the draft and for recommending publication to scipost. Below, we respond to his/her comments.

Requested changes Specific comments:

  1. On multiple occasions in the draft, calling “amplitude” of realizing the Rydberg excitation at any site as “detuning” is very misleading, especially from an atom-light interaction point of view. Detuning has units of frequency (equivalently energy) and well-defined interpretation even without any Rydberg excitations.

    Response: It is certainly true that "detuning" in atomic physics has well-defined interpretation even without invoking Rydberg excitations. However, in the field of Rydberg atoms the word detuning is very commonly used to mean the amplitude of Rydberg atoms. This is due to the fact that this amplitude depends on the difference of the laser frequency responsible for the excitation with the energy level separating the ground and excited Rydberg states. For details, we request the referee to look at Refs 25-28. We also note that we have already provided a clarification of this terminology in the introduction " The amplitude of realizing a Rydberg excitations at any site of such a chain, termed as detuning, can be individually controlled using external Raman lasers [25–28]".

  2. One of the essential ingredients of the model is the staggered detuning; it would be beneficial if the authors could provide a scheme to implement such a pattern in detuning along the ladder. In any case, it requires local manipulations using other optical fields and may not be as straightforward as the authors think. Along these lines, the papers cited in the introduction, Refs [50-59] do not provide how to implement that either.

    Response: It is indeed not straightforward to implement a staggered detuning in experiments; in fact these experiments for Rydberg chains have not been performed yet. However, we feel that the theoretical predictions about systems with such staggered detuning are rich enough [ Refs 45-54] so that such experiments may happen in future. We agree with the referee that local manipulation of the optical field required is not straightforward. However some suggestion regarding addressing detuning of individual Rydberg atoms in an array can be found in the literature (for example, see arXiv:2102.00341); we have now included this paper and also mentioned explicitly that staggered detuning are not easily achieved yet in the discussion section section of the paper.

  3. Note that quantum order by disorder transition is also discussed using a Rydberg + square lattice setup in https://journals.aps.org/prx/pdf/10.1103/PhysRevX.4.041037 but in a completely different interaction scheme.

    Response: This is indeed the case. We thank the referee for pointing out this paper. However, to the best of our knowledge, this paper does not explore the possible phases and the possible quantum phase transition between these phases of the order-by-disorder Hamiltonian constructed in Sec V. This is left as a subject of future study. For example in Sec V of that paper, the authors clearly remark " This setup might then constitute a perfect setting for the investigation of the competition between different RVB solid orders and the transitions between them" [ by setup, they mean Hamiltonian constructed in Sec V].

  4. In Eq. 1, the Hamiltonian for the Rydberg ladder, the Rabi term or the transverse magnetic field term (\sigma_x) has an additional negative sign, in contrast to the Hamiltonians describing the original Rydberg setups. It doesn’t look like the sign matters for the Physics discussed in the manuscript, as in the effective Hamiltonian in Eq. 17, it is w^2 in the first term, but it is better to clarify it.

    Response: The referee is correct that the choice of sign does not matter here. We have clarified this in the present version in Sec 2 after Eq 1.

  5. When the exact 1/r^6 interaction is considered, the transition point shifts to a larger value of \lambda, and a further increase in lambda makes a jump in the Rydberg density to n=1/2. Then, mentioning lambda>lambda_c, the system chooses a single classical configuration in the presence of weak quantum fluctuations arising from the Rabi coupling. Is not better to say, lambda_c < lambda < lambda_u, where lambda_u defines the upper value of n=1/4 plateau, the system chooses a single configuration?

    Response: We thank the referee for this suggestion and we have mentioned in Sec 4 now that $\lambda_c <\lambda_u for our chosen range of $w$ in the main text. However, we note that between for lambda_c < lambda < lambda_u, the ground state is doubly degenerate (single configuration may be chosen due to spontaneous symmetry breaking, if that's what the referee means).

  6. If we do an ED with the model (1) and truncate to nearest neighbour interaction, do we get the critical lambda=0.33 Delta?

    Response: Yes, it does.

  7. It is also good to comment on what happens if the Rabi coupling becomes sufficiently large. The ground state will have contributions from the degenerate manifold of the first excited state in Fig. 5d.

    Response: We thank the referee for this comment and have discussed it in the paper. We note that when the van der Waals interaction is curtailed to the nearest-neighbor only, $w$ is responsible only in determining the width of the plateau where the ground state manifold is degenerate; thus a sufficiently large w will ultimately destroy the transition via reduction of the width of the plateau (see Fig 2a). However, for realistic Rydberg interaction, one also needs a minimal value of $w$ below which $\lambda_c > \lambda_u$ [where \lambda_u>0 denotes the edge of the plateau] and the transition does not take place. Along with this, of course, there is also a cutoff for upper value of w where contribution from excited states ( as shown in Fig 5d) will obliterate the O(L) near-degeneracy and destroy the transition as discussed in Sec 4 now. Fortunately our numerics finds that there is indeed a window of values of $w$ where the transition is possible. We have put in a comment regarding the effect of large $w$ in the paragraph following Eq 12.

Anonymous on 2022-08-26  [id 2762]

(in reply to Krishnendu Sengupta on 2022-08-09 [id 2717])
Category:
remark
correction

This comment does not question the main results in the draft.
It is hard to agree with the authors that " in the field of Rydberg atoms, the word detuning is very commonly used to mean the amplitude of Rydberg atoms".

Detuning is the difference in energy between the light frequency and the energy level separation. The amplitude of Rydberg excitation has a non-trivial defence on detuning, Rabi frequency and Rydberg-Rydberg interactions.
The authors probably have in mind that, by increasing the detuning, one can get more Rydberg excitations in the lattice due to the anti-blockade, assuming the Rydberg-Rydberg interaction is repulsive. So the amplitude of Rydberg excitations can be controlled using the detuning of the excitation laser.

It is also instructive for authors to carefully look at the original dynamical crystallization paper: https://journals.aps.org/prl/pdf/10.1103/PhysRevLett.104.043002, which is very relevant to their work.
And also the first experimental work: https://www.science.org/doi/abs/10.1126/science.1258351.

As the authors comment: "The amplitude of realizing a Rydberg excitations at any site of such a chain, termed as detuning, can be individually controlled using external Raman lasers."

It is also unclear why they say Raman lasers unless the Rydberg excitation scheme involves a Raman transition. Typically it's a two-photon transition.

---

## Round 1 · Referee Report · Anonymous (Referee 2) · 2022-7-17

Strengths

  1. The system proposed realizes examples of quantum phase transitions arising from an order-by-disorder mechanism.
  2. The nature of the phase transition is examined both numerically and analytically.

Weaknesses

  1. The study does not fully elucidate the reason why the effective model Eq. 17 well explains the critical detuning obtained from ED.

Report

The paper studies the ground-state phase diagram and quantum phase transitions in a class of Rydberg ladders with uniform and staggered detunings. When the uniform detuning $\lambda$ is negatively large, the ground state does not have any Rydberg excitation. On the other hand, when $\lambda$ is positively large, the ground state is two-fold degenerate and maximizes the number of possible Rydberg excitations. In between, the ground state of the classical part is exponentially degenerate. However, quantum fluctuations lift this degeneracy and lead to a ground state with a definite order, which is called an order-by-disorder mechanism. To investigate the nature of the system, the authors first study the phases of the effective Hamiltonian given by Eq. 6. Then they demonstrate that a simple variational calculation based on the above physical picture well explains the ED results and argue that the phase transition on the $n=1/4$ plateau belongs to the 2D Ising universality class. This is further justified by examining the effective Hamiltonian Eq. 17. In Sec. 4, the effect of the long-range van der Waals interaction is discused numerically and analytically. The authors conclude that the further neighbor interactions do not obliterate the Ising transition, although they shift the transition to a higher value of $\lambda/\Delta$. The models on $n$-leg ladders are discussed in Appendices.

The manuscript is overall well written and nicely illustrates how this seemingly less obvious phase transition occurs in the system. In addition, the subject is certainly timely considering the recent research interest in Rydberg-atom systems. Thus I think the manuscript is well suited for publication in SciPost Physics. Nevertheless, I would like to suggest that the authors address the following point before publication:

The authors emphasize that a simple estimate based on a variational excited state for the effective Hamiltonian for $H'$ yields the approximate transition point, which is remarkably close to the one obtained from ED. However, this may not be a big mystery because the value of the transition point $\lambda_c = \Delta/3$ is exact for $H'$. This can be seen by mapping $H'$ in Eq. 17 to a transverse-field Ising chain, which is achieved by defining an effective spin-1/2 degree of freedom for each even rung. (Since the spin configurations on odd rungs are totally frozen, we can ignore them in Eq. 17.) With the identification $|\uparrow_{2j,1}, \downarrow_{2j,2}\rangle$ <-> $|\Uparrow_{2j}\rangle$ and $|\downarrow_{2j,1}, \uparrow_{2j,2}\rangle$ <-> $|\Downarrow_{2j}\rangle$, one can rewrite Eq. 17 as a Hamiltonian of the transverse field Ising chain, where the first and second terms correspond to the Ising-interaction and transverse-field terms, respectively, for the effective spins $|\Uparrow\rangle$ and $|\Downarrow\rangle$. Then it follows from the Kramers-Wannier duality that the transition point is exactly at $\alpha_0 = 2$, leading to $\lambda_c = \Delta/3$. The same story applies to the effective Hamiltonian of the 3-leg ladder model (Eq. 34). One can deduce the exact value of the transition point $\lambda_c = \Delta/2$ from the known result for the 3-state quantum Potts chain. It would be beneficial to the reader if the authors could include these discussions in the revision.

Requested changes

Minor comments: 1. In Eq. 1, the first sum is written as $\sum_{j=1,2L}$. However, it is less ambiguous to write the sum as $\sum_{j=1}^{2L}$. The same applies in numerous places in the draft like Eq. 5, 6, ...

  1. The argument around Eq. 5 is confusing, as $H_0$ pops up suddenly. I guess $H_0$ there means $H_a$ in Eq. 4.

  2. In Fig.2, the variational and ED results are shown for comparison. But what is the system size for the ED results? (In the main text, it is just said that $N \le 40$.)

  3. I think Eq. 18 should be replaced with the superposition of all Fock states with one Rydberg excitation. The current one is a superposition of just two states. (There may be a better way to express this excited state using an operator acting on the ground state $|\psi_G\rangle$ defined below Eq. 18.) A similar problem happens in Eq. 36 in Appendix B.

  4. The $\ell_0$-state quantum Potts model exhibits a first-order phase transition for $\ell_0 > 4$. Thus it is quite natural to expect that the same applies to the current $\ell_0$-leg ladder models. I wonder if the authors can comment on this. (I suggest that the authors at least remark that the nature of the phase transition of the Potts model depends on the value of $\ell_0$.)

  • validity: high
  • significance: good
  • originality: high
  • clarity: high
  • formatting: excellent
  • grammar: excellent

Author:  Krishnendu Sengupta  on 2022-08-09  [id 2718]

(in reply to Report 2 on 2022-07-17)

The paper studies the ground-state phase diagram and quantum phase transitions in a class of Rydberg ladders with uniform and staggered detunings. When the uniform detuning λ is negatively large, the ground state does not have any Rydberg excitation. On the other hand, when λ is positively large, the ground state is two-fold degenerate and maximizes the number of possible Rydberg excitations. In between, the ground state of the classical part is exponentially degenerate. However, quantum fluctuations lift this degeneracy and lead to a ground state with a definite order, which is called an order-by-disorder mechanism. To investigate the nature of the system, the authors first study the phases of the effective Hamiltonian given by Eq. 6. Then they demonstrate that a simple variational calculation based on the above physical picture well explains the ED results and argue that the phase transition on the n=1/4 plateau belongs to the 2D Ising universality class. This is further justified by examining the effective Hamiltonian Eq. 17. In Sec. 4, the effect of the long-range van der Waals interaction is discussed numerically and analytically. The authors conclude that the further neighbor interactions do not obliterate the Ising transition, although they shift the transition to a higher value of λ/Δ. The models on n-leg ladders are discussed in Appendices.

The manuscript is overall well written and nicely illustrates how this seemingly less obvious phase transition occurs in the system. In addition, the subject is certainly timely considering the recent research interest in Rydberg-atom systems. Thus I think the manuscript is well suited for publication in SciPost Physics. Nevertheless, I would like to suggest that the authors address the following point before publication:

Response: We thank the referee for a careful reading of the manuscript.

The authors emphasize that a simple estimate based on a variational excited state for the effective Hamiltonian for H′ yields the approximate transition point, which is remarkably close to the one obtained from ED. However, this may not be a big mystery because the value of the transition point λ_c =Δ/3 is exact for H′. This can be seen by mapping H′ in Eq. 17 to a transverse-field Ising chain, which is achieved by defining an effective spin-1/2 degree of freedom for each even rung. (Since the spin configurations on odd rungs are totally frozen, we can ignore them in Eq. 17.) With the identification |↑{2j,1},↓⟩<-> |⇑{2j}⟩ and |↓,↑{2j,2}⟩ <-> |⇓⟩, one can rewrite Eq. 17 as a Hamiltonian of the transverse field Ising chain, where the first and second terms correspond to the Ising-interaction and transverse-field terms, respectively, for the effective spins |⇑⟩ and |⇓⟩. Then it follows from the Kramers-Wannier duality that the transition point is exactly at α_0=2, leading to λ_c=Δ/3. The same story applies to the effective Hamiltonian of the 3-leg ladder model (Eq. 34). One can deduce the exact value of the transition point λ_c=Δ/2 from the known result for the 3-state quantum Potts chain. It would be beneficial to the reader if the authors could include these discussions in the revision.

Response: We thank the referee for this comment and we have included this discussion in the draft in Sec 3. A slightly modified argument in the same line as suggested by the referee indicates that the transition occurs at $\alpha_0=1/2$ which yields $\lambda_c=\Delta/3$.

Requested changes Minor comments: 1. In Eq. 1, the first sum is written as ∑{j=1,2L}. However, it is less ambiguous to write the sum as ∑^{2L}. The same applies in numerous places in the draft like Eq. 5, 6, ...

Response: We have now changed this in the present version.
  1. The argument around Eq. 5 is confusing, as H_0 pops up suddenly. I guess H_0 there means H_a in Eq. 4.

    Response: We have replaced H_0 by H_a which is what it should be.

  2. In Fig.2, the variational and ED results are shown for comparison. But what is the system size for the ED results? (In the main text, it is just said that N≤40.)

    Response: We have now mentioned the system size(N=20) in the caption of Fig 2 of the paper. We note that large values of N does not change the plot.

  3. I think Eq. 18 should be replaced with the superposition of all Fock states with one Rydberg excitation. The current one is a superposition of just two states. (There may be a better way to express this excited state using an operator acting on the ground state |ψ_G⟩ defined below Eq. 18.) A similar problem happens in Eq. 36 in Appendix B.

    Response: In our case, the excitation, once created, remains localized on the pair of sites where it is created; it does not have any kinetic term to O(w^2) in perturbation theory. This is in contrast to standard particle/hole excitation in the Bose-Hubbard model or domain wall/flipped spin excitation in the FM/PM phase of the standard Ising model. Consequently, a creation of superposition of such excited states over different sites of the ladder does not change the result within O(w^2). This is also why the variational state chosen provides remarkably accurate results.

  4. The ℓ_0-state quantum Potts model exhibits a first-order phase transition for ℓ_0>4. Thus it is quite natural to expect that the same applies to the current ℓ_0-leg ladder models. I wonder if the authors can comment on this. (I suggest that the authors at least remark that the nature of the phase transition of the Potts model depends on the value of ℓ_0.)

    Response: We have made a comment about this point in the draft in the appendix, where we discuss the transition mentioning that for $\ell_0 \ge 5$, we expect a first order transition. However our numerics can not reach large enough system sizes where this could be numerically probed. Also, our discussion following Eq 37 ties the universality with $\ell_0$: thus the fact that for $\ell_0=3$, the transition will belong to 3-state Potts is clear from there.

---

## Round 2 · Referee Report · Anonymous (Referee 2) · 2022-9-4

Report

I have gone through the revised manuscript and the authors' response to my comments. They addressed all the comments and improved the quality/readability of the paper accordingly. Concerning my main suggestion, the authors are indeed correct. I made a silly mistake when I wrote my previous report, and the correct $\alpha_0$ leading to $\lambda_c=\Delta/3$ should be $1/2$. In any case, I was happy to see that my suggestion was helpful.

I think the current manuscript is almost ready for acceptance. However, there is concern about the use of the term "detuning." As the other referee says, the word detuning means the difference in energy between the light frequency and the energy level separation. This also applies in the field of Rydberg atoms. Thus, I would suggest the authors should follow the suggestion and use more careful wording in the revision.
  • validity: -
  • significance: -
  • originality: -
  • clarity: -
  • formatting: -
  • grammar: -

Author:  Krishnendu Sengupta  on 2022-09-13  [id 2807]

(in reply to Report 1 on 2022-09-04)

Ref: I have gone through the revised manuscript and the authors' response to my comments. They addressed all the comments and improved the quality/readability of the paper accordingly. Concerning my main suggestion, the authors are indeed correct. I made a silly mistake when I wrote my previous report, and the correct α0 leading to λc=Δ/3 should be 1/2. In any case, I was happy to see that my suggestion was helpful.

Response: We thank the referee for this suggestion which was indeed helpful.

Referee : I think the current manuscript is almost ready for acceptance. However, there is concern about the use of the term "detuning." As the other referee says, the word detuning means the difference in energy between the light frequency and the energy level separation. This also applies in the field of Rydberg atoms. Thus, I would suggest the authors should follow the suggestion and use more careful wording in the revision.

Regarding the comment on detuning we would like to point out the following:

a) We understand that detuning means the energy difference between the light frequency and energy level separation; it is denoted by $\Delta$. It is well known ( see Refs 25 and 26 of our work and the discussion after Eq 1 in both the work) that this detuning is the parameter that enters the effective Hamiltonian of the Rydberg atoms.

b) We would like to point out that the above-mentioned identification has been widely used recently in both theoretical and experimental literature. (See Refs 25, 26, 37, 45, 46).

c) We agree that one has two-photon transition here; so we replace Raman lasers by external lasers.

d) To make things clear, we have now explicitly mentioned on Pg 1 para 2

"The amplitude of realizing a Rydberg
excitation at any site of such a chain can be controlled by changing the detuning, i.e, the difference
between the energy (frequency) of the external laser and the energy gap between the ground and
excited states of a Rydberg atom"

and changed (in Sec 2 below Eq 1)

" Here λ and ∆ denote the amplitudes of uniform and staggered detuning respectively, and w > 0 denotes the coupling strength between the Rydberg ground and excited states."

to

In Eq. 1, λ and ∆ denote the amplitudes of uniform and staggered
detuning such that λ + ∆ and λ − ∆ represent the energy differences between the energy of the
applied external laser and the energy gap between ground and excited Rydberg atomic levels on
even and odd sites respectively. Here w > 0 denotes the coupling strength between the Rydberg
ground and excited states. In an experimental setup, this coupling is controlled by two-photon
processes having Rabi frequency w/ħ [25–28]."

in the draft.

We hope this will make our notation clear. We have now appropriately changed the expressions of our Hamiltonian to change the sign of the term \sim $w$ so that $w$ can be identified with the Rabi frequency (we have already noted earlier that such a change of sign does not alter any result).

---

## Round 2 · Author Response

Response to referee comments:

Response to the second referee:

The paper studies the ground-state phase diagram and quantum phase transitions in a class of Rydberg ladders with uniform and staggered detunings. When the uniform detuning λ is negatively large, the ground state does not have any Rydberg excitation. On the other hand, when λ is positively large, the ground state is two-fold degenerate and maximizes the number of possible Rydberg excitations. In between, the ground state of the classical part is exponentially degenerate. However, quantum fluctuations lift this degeneracy and lead to a ground state with a definite order, which is called an order-by-disorder mechanism. To investigate the nature of the system, the authors first study the phases of the effective Hamiltonian given by Eq. 6. Then they demonstrate that a simple variational calculation based on the above physical picture well explains the ED results and argue that the phase transition on the n=1/4 plateau belongs to the 2D Ising universality class. This is further justified by examining the effective Hamiltonian Eq. 17. In Sec. 4, the effect of the long-range van der Waals interaction is discussed numerically and analytically. The authors conclude that the further neighbor interactions do not obliterate the Ising transition, although they shift the transition to a higher value of λ/Δ. The models on n-leg ladders are discussed in Appendices.

The manuscript is overall well written and nicely illustrates how this seemingly less obvious phase transition occurs in the system. In addition, the subject is certainly timely considering the recent research interest in Rydberg-atom systems. Thus I think the manuscript is well suited for publication in SciPost Physics. Nevertheless, I would like to suggest that the authors address the following point before publication:

Response: We thank the referee for a careful reading of the manuscript.

The authors emphasize that a simple estimate based on a variational excited state for the effective Hamiltonian for H′ yields the approximate transition point, which is remarkably close to the one obtained from ED. However, this may not be a big mystery because the value of the transition point λ_c=Δ/3 is exact for H′. This can be seen by mapping H′ in Eq. 17 to a transverse-field Ising chain, which is achieved by defining an effective spin-1/2 degree of freedom for each even rung. (Since the spin configurations on odd rungs are totally frozen, we can ignore them in Eq. 17.) With the identification |↑{2j,1},↓{2j, 2⟩ <-> |⇑{2j⟩ and |↓,↑{2j,2⟩ <-> |⇓⟩ , one can rewrite Eq. 17 as a Hamiltonian of the transverse field Ising chain, where the first and second terms correspond to the Ising-interaction and transverse-field terms, respectively, for the effective spins |⇑⟩ and |⇓⟩. Then it follows from the Kramers-Wannier duality that the transition point is exactly at α_0=2, leading to λ_c=Δ/3. The same story applies to the effective Hamiltonian of the 3-leg ladder model (Eq. 34). One can deduce the exact value of the transition point λ_c=Δ/2 from the known result for the 3-state quantum Potts chain. It would be beneficial to the reader if the authors could include these discussions in the revision.

Response: We thank the referee for this comment and we have included this discussion in the draft in Sec 3. We find that a slightly more careful mapping in line of suggestion by the referee leads to the transition point at \alpha_0 =1/2 for the two-leg ladder ( this gives the correct value of lambda_c= \Delta/3).

Requested changes Minor comments: 1. In Eq. 1, the first sum is written as ∑{j=1,2L} . However, it is less ambiguous to write the sum as ∑^{2L}. The same applies in numerous places in the draft like Eq. 5, 6, ...

Response: We have now changed this in the present version.

  1. The argument around Eq. 5 is confusing, as H_0 pops up suddenly. I guess H_0 there means H_a in Eq. 4.

Response: We have replaced H_0 by H_a which is what it should be.

  1. In Fig.2, the variational and ED results are shown for comparison. But what is the system size for the ED results? (In the main text, it is just said that N ≤40.)

Response: We have now mentioned the system size in the caption of Fig 2 of the paper ( it corresponds to N=20). We find that the plot does not change with increasing N.

  1. I think Eq. 18 should be replaced with the superposition of all Fock states with one Rydberg excitation. The current one is a superposition of just two states. (There may be a better way to express this excited state using an operator acting on the ground state |ψ_G⟩ defined below Eq. 18.) A similar problem happens in Eq. 36 in Appendix B.

Response: In our case, the excitation, once created, remains localized on the pair of sites where it is created; it does not have any kinetic term to O(w^2) in perturbation theory. This is in contrast to standard particle/hole excitation in the Bose-Hubbard model or domain wall/flipped spin excitation in the FM/PM phase of the standard Ising model. Consequently, a creation of superposition of such excited states over different sites of the ladder does not change the result within O(w^2). This is also why the variational state chosen provides remarkably accurate results.

  1. The ℓ_0 -state quantum Potts model exhibits a first-order phase transition for ℓ_0>4. Thus it is quite natural to expect that the same applies to the current ℓ_0-leg ladder models. I wonder if the authors can comment on this. (I suggest that the authors at least remark that the nature of the phase transition of the Potts model depends on the value of ℓ_0.)

Response: We have made a comment about this point in the draft in the appendix, where we discuss the transition mentioning that for $\ell_0 \ge 5$, we expect a first order transition. However our numerics can not reach large enough system sizes where this could be numerically probed. Also, our discussion following Eq 37 ties the universality with $\ell_0$: thus the fact that for $\ell_0=3$, the transition will belong to 3-state Potts is clear from there.

&&&&&&&&&&&&&&&&&&&&&&&&&&&&&&&&&&&&&&&&&&&&&&&&&&&&

First Referee:

Referee: The authors studied the ground state configurations in a Rydberg ladder in the presence of both staggered (Delta) and uniform (lambda) detunings. In the case of the nearest neighbour blockade, the system exhibits macroscopic degenerate states with a Rydberg filling of n=1/4 for a finite range of lambda/Delta ratio. Within the n=1/4 plateau, above a critical value of uniform detuning, the system chooses one of the classical configurations in which Rydberg atoms are excited only in one of the chains, breaking the Z2 symmetry via quantum order by disorder mechanism. In Sec. 3, the physics is analyzed using effective Hamiltonians. In contrast, in Sec. 4, the original Hamiltonian for a Rydberg ladder is used and shows the existence of the critical point in the n=1/4 plateau. The quantum order by disorder transition emerges from the staggered detuning and the Rabi coupling. Finally, the study is extended beyond the two-leg ladder scheme. In particular, the three-leg ladder possesses a quantum critical point with a three-states Pott universality class in 2D. This work is exciting; at first, it was less obvious, but later it became more apparent in the draft. I recommend the paper to publish in the scipost physics. More specific comments are provided below.

Response: We thank the referee for a careful reading of the draft and for recommending publication to scipost. Below, we respond to his/her comments.

Requested changes Specific comments:

  1. On multiple occasions in the draft, calling “amplitude” of realizing the Rydberg excitation at any site as “detuning” is very misleading, especially from an atom-light interaction point of view. Detuning has units of frequency (equivalently energy) and well-defined interpretation even without any Rydberg excitations.

Response: It is certainly true that "detuning" in atomic physics has well-defined interpretation even without invoking Rydberg excitations. However, in the field of Rydberg atoms the word detuning is very commonly used to mean the amplitude of Rydberg atoms. This is due to the fact that this amplitude depends on the difference of the laser frequency responsible for the excitation with the energy level separating the ground and excited Rydberg states. For details, we request the referee to look at Refs 25-28. We also note that we have already provided a clarification of this terminology in the introduction " The amplitude of realizing a Rydberg excitations at any site of such a chain, termed as detuning, can be individually controlled using external Raman lasers [25–28]".

  1. One of the essential ingredients of the model is the staggered detuning; it would be beneficial if the authors could provide a scheme to implement such a pattern in detuning along the ladder. In any case, it requires local manipulations using other optical fields and may not be as straightforward as the authors think. Along these lines, the papers cited in the introduction, Refs [50-59] do not provide how to implement that either.

Response: It is indeed not straightforward to implement a staggered detuning in experiments; in fact these experiments for Rydberg chains have not been performed yet. However, we feel that the theoretical predictions about systems with such staggered detuning are rich enough [ Refs 45-54] so that such experiments may happen in future. We agree with the referee that local manipulation of the optical field required is not straightforward. However some suggestion regarding addressing detuning of individual Rydberg atoms in an array can be found in the literature (for example, see arXiv:2102.00341); we have now included this paper and also mentioned explicitly that staggered detuning are not easily achieved yet in the discussion section section of the paper.

  1. Note that quantum order by disorder transition is also discussed using a Rydberg + square lattice setup in https://journals.aps.org/prx/pdf/10.1103/PhysRevX.4.041037 but in a completely different interaction scheme.

Response: This is indeed the case. We thank the referee for pointing out this paper. However, to the best of our knowledge, this paper does not explore the possible phases and the possible quantum phase transition between these phases of the order-by-disorder Hamiltonian constructed in Sec V. This is left as a subject of future study. For example in Sec V of that paper, the authors clearly remark " This setup might then constitute a perfect setting for the investigation of the competition between different RVB solid orders and the transitions between them" [ by setup, they mean Hamiltonian constructed in Sec V].

  1. In Eq. 1, the Hamiltonian for the Rydberg ladder, the Rabi term or the transverse magnetic field term (\sigma_x) has an additional negative sign, in contrast to the Hamiltonians describing the original Rydberg setups. It doesn’t look like the sign matters for the Physics discussed in the manuscript, as in the effective Hamiltonian in Eq. 17, it is w^2 in the first term, but it is better to clarify it.

Response: The referee is correct that the choice of sign does not matter here. We have clarified this in the present version in Sec 2 after Eq 1.

  1. When the exact 1/r^6 interaction is considered, the transition point shifts to a larger value of \lambda, and a further increase in lambda makes a jump in the Rydberg density to n=1/2. Then, mentioning lambda>lambda_c, the system chooses a single classical configuration in the presence of weak quantum fluctuations arising from the Rabi coupling. Is not better to say, lambda_c < lambda < lambda_u, where lambda_u defines the upper value of n=1/4 plateau, the system chooses a single configuration?

Response: We thank the referee for this suggestion and we have mentioned in Sec 4 now that $\lambda_c <\lambda_u for our chosen range of $w$ in the main text. However, we note that between for lambda_c < lambda < lambda_u, the ground state is doubly degenerate (single configuration may be chosen due to spontaneous symmetry breaking, if that's what the referee means).

  1. If we do an ED with the model (1) and truncate to nearest neighbour interaction, do we get the critical lambda=0.33 Delta?

Response: Yes, it does.

  1. It is also good to comment on what happens if the Rabi coupling becomes sufficiently large. The ground state will have contributions from the degenerate manifold of the first excited state in Fig. 5d.

Response: We thank the referee for this comment and have discussed it in the paper. We note that when the van der Waals interaction is curtailed to the nearest-neighbor only, $w$ is responsible only in determining the width of the plateau where the ground state manifold is degenerate; thus a sufficiently large w will ultimately destroy the transition via reduction of the width of the plateau (see Fig 2a). However, for realistic Rydberg interaction, one also needs a minimal value of $w$ below which $\lambda_c > \lambda_u$ [where \lambda_u>0 denotes the edge of the plateau] and the transition does not take place. Along with this, of course, there is also a cutoff for upper value of w where contribution from excited states ( as shown in Fig 5d) will obliterate the O(L) near-degeneracy and destroy the transition as discussed in Sec 4 now. Fortunately our numerics finds that there is indeed a window of values of $w$ where the transition is possible. We have put in a comment regarding the effect of large $w$ in the paragraph following Eq 12.

&&&&&&&&&&&&&&&&&&&&&&&&&&&&&&&&&&&&&&&&&&&&&&&

---

## Round 2 · List of Changes

List of changes:

1) We have added system size dependence (N=20) in the caption of Fig 2.

2) We have added a discussion on mapping of the effective Hamiltonian for two chains to an Ising model in transverse field at the end of Sec 3.

3) We have changed the summation convention as per request of the second referee.

4) We have replaced $H_0$ by $H_a$ in text around Eq 5.

5) We have added a comment on the nature of the transition for $\ell_0 >4$ in App B.

6) We have now added a comment regarding possibility of realization of staggered detuning in these chains. We have also added a reference ( Ref 66) regarding this.

7) We have commented about irrelevance of sign of w in our work in Sec 2 after Eq 1.

8) We have comment about the range of $w$ for which the transition is seen ( in the presence of finite higher-neighboring terms of vdW interaction) in Sec 4 of the main text in response to points 5 and 7 of the first referee.

---

## Editorial Decision

resubmitted